# Epigenetic reprogramming rewires transcription during the alternation of generations in Arabidopsis

Michael Borg[1], Ranjith K Papareddy[1], Rodolphe Dombey[1], Elin Axelsson[1], Michael D Nodine[1], David Twell[1,2], Frédéric Berger[1]*

[1]Gregor Mendel Institute (GMI), Austrian Academy of Sciences, Vienna, Austria; [2]Department of Genetics, University of Leicester, Leicester, United Kingdom

**Abstract** Alternation between morphologically distinct haploid and diploid life forms is a defining feature of most plant and algal life cycles, yet the underlying molecular mechanisms that govern these transitions remain unclear. Here, we explore the dynamic relationship between chromatin accessibility and epigenetic modifications during life form transitions in Arabidopsis. The diploid-to-haploid life form transition is governed by the loss of H3K9me2 and DNA demethylation of transposon-associated *cis*-regulatory elements. This event is associated with dramatic changes in chromatin accessibility and transcriptional reprogramming. In contrast, the global loss of H3K27me3 in the haploid form shapes a chromatin accessibility landscape that is poised to re-initiate the transition back to diploid life after fertilisation. Hence, distinct epigenetic reprogramming events rewire transcription through major reorganisation of the regulatory epigenome to guide the alternation of generations in flowering plants.

*For correspondence:
Frederic.berger@gmi.oeaw.ac.at

**Competing interests:** The authors declare that no competing interests exist.

## Introduction

Epigenetic reprogramming refers to the global erasure and remodelling of epigenetic marks during development. DNA and histone methylation are reprogrammed during germline differentiation and after fertilisation to reconfigure transcription in the mammalian embryo (*Hajkova, 2011*; *Reik et al., 2001*). In plants and algae, meiosis marks the transition from the diploid sporophyte to haploid gamete-producing gametophytes, while the union of gametes at fertilisation re-initiates the transition back to the diploid phase (*Hackenberg and Twell, 2019*; *Haig, 2008*; *Hisanaga et al., 2019*). The marked differences in development, morphology, and gene expression between the gametophyte and sporophyte (*Borg et al., 2009*; *Couceiro et al., 2015*; *Dickinson and Grant-Downton, 2009*; *Heesch et al., 2019*) suggest that epigenetic patterns are reprogrammed between these two life forms. Nevertheless, the nature of epigenetic reprogramming and its influence on transcriptional activity during the alternation of generations remain poorly understood.

Ancestral land plants represented by extant bryophytes typically have a dominant haploid gametophytic phase and require Polycomb Repressive Complex 2 (PRC2) activity to repress precocious sporophyte development (*Okano et al., 2009*). DNA methylation also undergoes extensive reprogramming between the haploid and diploid phase in bryophytes, although it remains unclear whether this is required for the transitions between the two life forms (*Ikeda et al., 2018*; *Schmid et al., 2018*; *Yaari et al., 2015*). In contrast, flowering plants have a dominant diploid sporophytic phase that culminates with the production of meiotic spores from somatic tissue within the flower. Haploid spores develop into multicellular male or female gametophytes depending on whether they originate in the anther or ovary, respectively (*Dickinson and Grant-Downton, 2009*). The female gametophyte in flowering plants is comprised of an embryo sac containing the egg and accessory cells (*Erbasol Serbes et al., 2019*). Pollen represents the male gametophyte. It develops from

**eLife digest** Each pollen grain from a flowering plant houses sperm, which contain half of the genes needed to make a new plant, and a companion or vegetative cell (VC) that serves to deliver sperm to the egg. The genes in the vegetative cell and those in the sperm are identical to the genes of the plant they come from, so how can this set of identical genetic information produce such different cells?

Both DNA and histones, the proteins that pack and order DNA, can be chemically modified locally through a process called methylation. The location of these modifications can affect how genetic information in the DNA is read to make different types of cells. The use of processes like methylation to regulate whether genes are switched on or off is called epigenetics. So what role does epigenetics play in plant pollen?

To answer this question, Borg et al. examined the epigenetics of pollen in *Arabidopsis thaliana*, a widely studied plant and common weed. In vegetative cells, DNA methylation is lost together with a different methylation mark (H3K9me2), which unlocks several genes needed for pollen to transport sperm. By contrast, sperm loses an entirely different methylation mark, called H3K27me3, which unlocks a different set of genes that help to prepare development of a new plant once sperm fertilizes the egg. Through these different set of epigenetic changes, activity increases at different groups of genes that are important for shaping the function of each pollen cell type.

These results reveal how the loss of DNA and histone methylation are important for plants to reproduce sexually via pollen. This offers insights into the evolution of plants and other related life forms. Learning about plant reproduction may also help to increase food production by improving crop yields.

asymmetric division of the haploid microspore, which produces a germ cell that becomes engulfed in the cytoplasm of its sister vegetative cell (VC) (*Figure 1A*; *Borg et al., 2009*). The germ cell represents the male germline initial cell and undergoes a distinct developmental programme to divide and differentiate into two sperms (*Borg et al., 2009*). Sperm differentiation is intricately linked with epigenetic reprogramming of sperm chromatin in Arabidopsis. Targeted removal of repressive H3K27me3 marks from histone-based sperm chromatin facilitates the transcription of genes required for sperm differentiation, which are normally silenced by H3K27me3 during sporophytic life (*Borg et al., 2020*).

In contrast to sperm, the events that govern differentiation of the VC remain largely unknown. VC differentiation involves acquiring the competency to grow a pollen tube for the delivery of sperm to the embryo sac, where fertilisation of the egg cell re-initiates diploid sporophyte development (*Hamamura et al., 2012*). Asymmetric microspore division is essential for pollen patterning since chemically induced symmetry results in two daughter cells with VC fate (*Eady et al., 1995*), suggesting that VC specification is the default developmental fate in the male gametophyte. VC nuclear (VN) chromatin organisation is highly diffuse compared to sperm nuclei (SN) and somatic cell nuclei (*Borg and Berger, 2015*). This is caused by the decondensation of pericentromeric heterochromatin, which involves the depletion of linker histone H1 and H3K9me2 (*He et al., 2019*; *Mérai et al., 2014*; *Schoft et al., 2009*; *Slotkin et al., 2009*). Heterochromatin decondensation licenses the active demethylation of pericentromeric DNA sequences by the gametophyte-specific DNA glycosylase DEMETER (DME) (*Andreuzza et al., 2010*; *He et al., 2019*; *Ibarra et al., 2012*; *Schoft et al., 2011*), which stimulates transcription of transposable elements (TEs) and epigenetically activated small RNA (easiRNA) (*Creasey et al., 2014*; *Slotkin et al., 2009*). Despite the large-scale nature of these epigenetic reconfigurations, it remains unclear if and how these might relate to differentiation of the VC, which ultimately embodies the paternal sporophyte-to-gametophyte transition in flowering plants.

Here, we assess chromatin and transcriptional reprogramming during the sporophyte-to-gametophyte transition in developing Arabidopsis pollen. We reveal widespread reprogramming of accessible chromatin across this transition and map hundreds of loci that gain accessibility in each pollen cell type. We demonstrate how the rewiring of *cis*-regulatory activity is intricately linked with the differential reprogramming of repressive epigenetic marks after microspore division. In the VC, accessible chromatin is shaped by the loss of constitutive heterochromatin and active cytosine

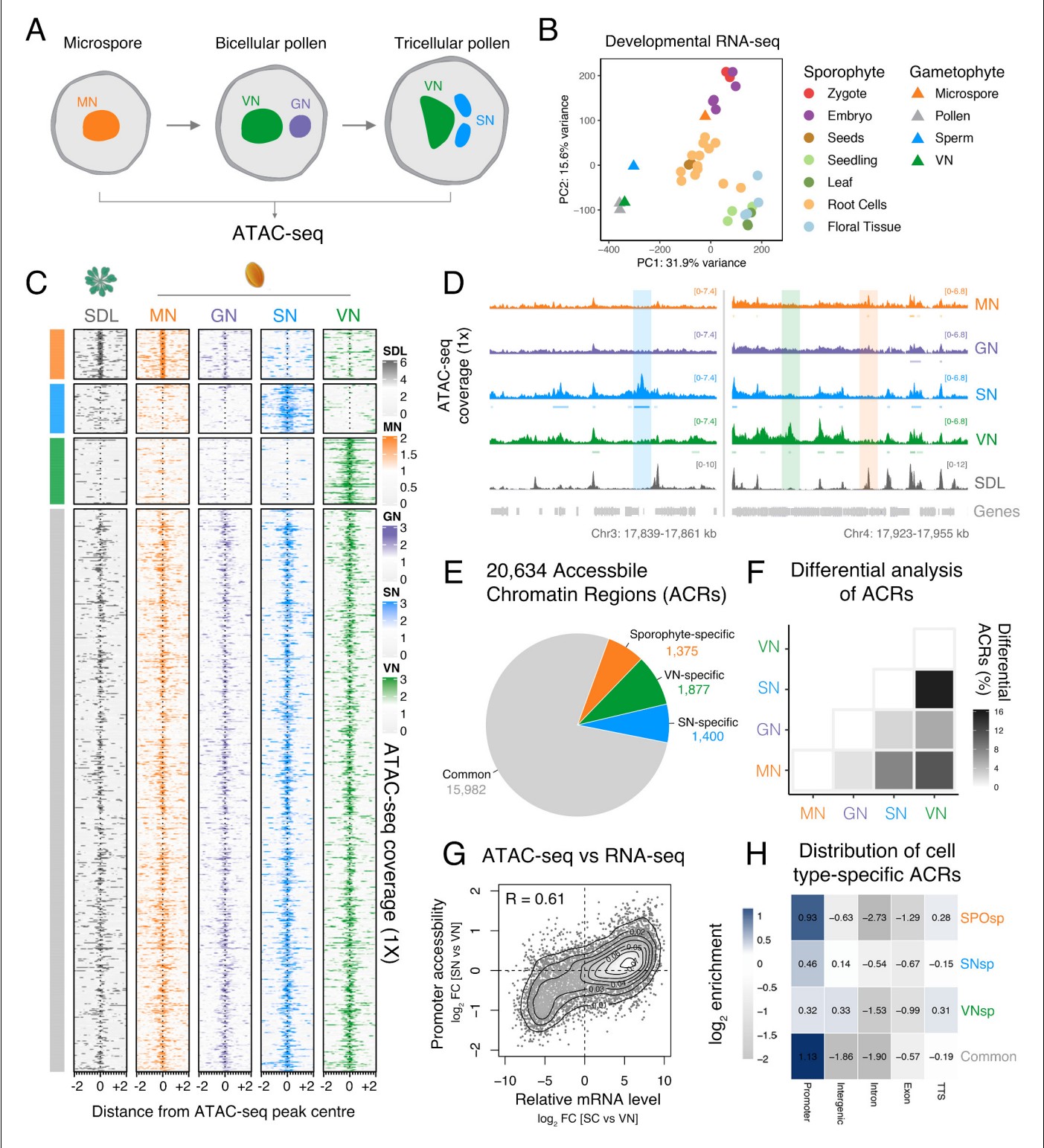

**Figure 1.** Chromatin accessibility is extensively reprogrammed in pollen. (**A**) Schematic of Arabidopsis pollen development. Microspores divide asymmetrically to produce a vegetative cell nucleus (VN, green) and germ cell nucleus (GN, purple). While the vegetative cell terminally differentiates, the germ cell divides to produce two sperm nuclei (SN, blue). Details of cellular membranes have been excluded for simplicity. ATAC-seq was performed on FACS-enriched microspores and on fluorescence-activated nuclear sorting (FANS)-isolated GN, SN, and VN. (**B**) Principal component analysis comparing Arabidopsis RNA-seq transcriptomes from sporophytic and male gametophytic cell types and tissues. Published RNA-seq data sets were reanalysed or described previously (*Borg et al., 2020*; *Hofmann et al., 2019*; *Nodine and Bartel, 2012*; *Wang et al., 2020*; *Zhao et al., 2019*).
*Figure 1 continued on next page*

*Figure 1 continued*

(C) Heatmap centred on pollen accessible chromatin regions (ACRs) comparing seedlings (SDL), microspore nuclei (MN), GN, SN, and VN. Clustering is based on ACR specificity to MN (orange), SN (blue), or VN (green). ACRs in common with seedlings are shown in grey. (D) Genome browser comparison of ATAC-seq tracks in MN, GN, SN, VN, and SDL. Examples of SN-specific (blue), VN-specific (green), and sporophyte and MN-specific (orange) ACRs are indicated with colour shading. ATAC-seq signal is normalised to 1× Arabidopsis genome coverage. (E) Pie chart summary of the specificity of pollen ACRs determined in this study. (F) Heatmap summarising the percentage of differential ACRs detected in pairwise comparisons between the MN, GN, SN, and VN. Differential accessibility was assessed within the 20,634 pollen ACRs shown in panel E. (G) Pairwise correlation of relative chromatin accessibly at promoters and relative mRNA levels of the associated transcribed genes between SN and VN. R represents Pearson's correlation coefficient. (H) Genomic distribution of ATAC-seq peaks in each different group of ACRs. The $\log_2$ enrichment or depletion is calculated relative to the frequency of genomic features in the Arabidopsis genome.

The online version of this article includes the following figure supplement(s) for figure 1:

**Figure supplement 1.** Isolation of key cell types across Arabidopsis pollen development.
**Figure supplement 2.** Transcriptomic profiling performed in this study.
**Figure supplement 3.** ATAC-seq profiling performed in this study.
**Figure supplement 4.** ATAC-seq profiling across Arabidopsis pollen development.

demethylation of *cis*-regulatory regions that are predicted to be bound by pollen-expressed transcription factors (TFs) involved in VC differentiation. In contrast, chromatin accessibility in sperm is shaped by the loss of H3K27me3, resulting in a chromatin state that forecasts the re-initiation of sporophyte development in the next generation. Our findings thus reveal how epigenetic reprogramming orchestrates the rewiring of transcription during haploid–diploid life form transitions in flowering plants.

## Results

### Pollen cell types have highly distinct transcriptional output

As sperm cells are embedded within the VC cytoplasm, existing Arabidopsis whole pollen transcriptomes represent a composite profile derived from both cell types (*Becker et al., 2003*; *Honys and Twell, 2004*). To circumvent this issue, we generated an RNA-seq profile from highly pure fractions of the VN (*Figure 1—figure supplement 1A and B*) and compared this with the transcriptome of isolated sperm (*Figure 1—figure supplement 2A–D*; *Borg et al., 2020*; *Borges et al., 2008*). Consistent with most pollen RNA deriving from the VC, the VN transcriptome strongly correlated with that of whole pollen but was poorly correlated with that of sperm (*Figure 1—figure supplement 2C and D*). This supported the VN transcriptome as a suitable proxy for VC-specific gene expression, as shown for nuclear transcriptomes from other cell types (*Deal and Henikoff, 2011*; *Slane et al., 2015*). A principal component analysis illustrates how the transcriptomes of two cells constituting mature pollen are highly distinct from each other and from transcriptomes of sporophytic cell types and tissues (*Figure 1B*). Gene expression in haploid microspores was more related to diploid tissues of the sporophyte than to the two differentiated pollen cell types. This indicates that major transcriptional reprogramming establishes distinct regulatory controls after unequal division of the microspore to specify the sperm and the VC.

### Mapping accessible chromatin during pollen development

To assess chromatin reprogramming in the male gametophyte, we profiled chromatin accessibility in the distinct cell types of the developing Arabidopsis pollen lineage using Assay for Transposase-Accessible Chromatin using sequencing (ATAC-seq) (*Buenrostro et al., 2013*; *Figure 1A*). We isolated highly pure fractions of SN and VN using a combination of cell-type specific markers and fluorescence-activated nuclear sorting (FANS) (*Figure 1—figure supplement 1A–D*), while microspores were isolated based on their size and auto-fluorescent properties (*Figure 1—figure supplement 1E–G*; *Borges et al., 2012*). To isolate germ cell nuclei (GN), we developed a tailored marker line that is active solely in the germ cell and performed FANS with nuclei isolated from developing spores (*Figure 1—figure supplement 1H and I*). For each cell type, we generated at least two ATAC-seq biological replicates and validated our profiles in several ways (*Figure 1—figure supplement 3A*). First, biological replicates for each stage were highly correlated, while modest cross-

correlation between tissues indicated distinct chromatin landscapes (*Figure 1—figure supplement 3B and C*). Second, ATAC-seq peaks were highly enriched at promoters (*Figure 1—figure supplement 4A*). Third, highly transcribed genes showed higher accessibility around transcription start sites (TSSs) than genes with low expression (*Figure 1—figure supplement 4B*). Fourth, ATAC-seq profiles in SN and VN had characteristic periodicity arising from polynucleosome fragments (*Figure 1—figure supplement 3D*). Although no periodicity was evident in profiles from microspore nuclei (MN) and GN (*Figure 1—figure supplement 3D*), we observed substantial overlap with profiles from other cell types as well as strong correlation of chromatin accessibility with active genes in the microspore and sperm, respectively (*Figure 1C and D*; *Figure 1—figure supplement 4B*). Our ATAC-seq profiles thus map distinct chromatin accessibility landscapes that span the inception and development of the male gametophyte.

## Chromatin accessibility is extensively reprogrammed in pollen

We first examined differences in chromatin accessibility between the microspore, sperm, and vegetative cell. We identified a total of 20,634 ATAC-seq peaks covering over 15 Mb of the Arabidopsis genome (*Figure 1C*; *Supplementary file 1*). We defined peak regions recovered by ATAC-seq as accessible chromatin regions (ACRs). Chromatin accessibility in microspores, differentiated pollen cell types, and seedlings strongly overlapped (15,982 peaks, 77.5%) (*Figure 1C–E*). These common peaks were associated with genes involved in core housekeeping functions that maintain expression across the entire life cycle, such as mRNA metabolism, protein transport, cell cycle control, and osmotic stress response (*Figure 1—figure supplement 4C*). A set of sporophyte-specific ACRs (6.7%; 1375 peaks) remained accessible between seedlings and MN, had reduced accessibility in the GN, but later lost accessibility in the SN and VN (*Figure 1C*). These occurred at genes involved in photosynthesis and energy metabolism (*Figure 1—figure supplement 4C*), which are dispensable in the short-lived haploid gametophyte since it derives resources from the host parental sporophyte. Our analysis did not reveal de novo gains in chromatin accessibility in GN compared with MN but rather a gradual loss in accessibility among sporophytic-specific ACRs (*Figure 1C*).

Pairwise comparisons between the pollen cell types revealed a gradual increase in differential accessibility across pollen development (*Figure 1F*; *Figure 1—figure supplement 3E*; *Supplementary file 1*). These differences were most pronounced at the mature pollen stage, with 6.8% (1400 peaks) and 9.1% (1877 peaks) cell type-specific ACRs unique to SN or VN, respectively (*Figure 1C–E*; *Supplementary file 1*). Heatmaps centred on ATAC-seq peaks illustrate the cell type-specificity of SN-specific and VN-specific ACRs (*Figure 1C*). VN-specific ACRs were strongly associated with functions unique to pollen biology such as pollination, cell tip growth, and morphogenesis (*Figure 1—figure supplement 4C*). In sperm, we observed high promoter accessibility at genes directly regulated by DUO POLLEN 1 (DUO1), an essential R2R3-type MYB TF that controls sperm differentiation (*Figure 1—figure supplement 4D*; *Borg et al., 2011*; *Brownfield et al., 2009*; *Rotman et al., 2005*). Importantly, the relative accessibility of promoter regions was positively correlated ($R$ = 0.61) with differential expression of the associated genes between sperm and the VN (*Figure 1G*), consistent with the distinct transcriptional state of each cell type. SN-specific and VN-specific ACRs were less prevalent at promoters but had an increased incidence within distal intergenic regions compared with common ACRs (*Figure 1H*). Transcribed genes most proximal to these intergenic ACRs were differentially expressed in a cell type-specific manner (*Figure 1—figure supplement 4E*), suggesting distal regulatory influence. VN-specific ACRs were also highly enriched at transcriptional termination sites (TTSs) compared to other ACRs (*Figure 1H*; *Figure 1—figure supplement 4B*). In summary, ATAC-seq has revealed hundreds of unique ACRs specific to the sperm or VC, which likely encompass putative *cis*-regulatory elements that modulate transcription and differentiation of each pollen cell type. These *cis*-regulatory regions are distributed between proximal promoters and distal regions, suggesting the involvement of long-range chromatin interactions in the regulation of pollen gene expression.

## Pericentromeric sequences gain chromatin accessibility in the male gametophyte

In Arabidopsis, condensed heterochromatin is organised into chromocenters and is comprised of pericentromeric sequences enriched in H3K9me2 and DNA methylation (*Feng and Michaels, 2015*).

These repressive epigenetic marks silence TEs that accumulate within pericentromeric regions (*Feng and Michaels, 2015*). Because pericentromeric heterochromatin is lost in the VN (*Schoft et al., 2009*), we explored how this might impact the landscape of accessible chromatin. ATAC-seq peak density was reduced within pericentromeric regions compared to the arms of each of the five Arabidopsis chromosomes (*Figure 2A*). We also noted a slight but consistent increase in ATAC-seq peak density within pericentromeric regions of the VN compared to SN (*Figure 2A*; *Figure 2—figure supplement 1A*). Consistent with this, and unlike common or SN-specific ACRs, VN-specific ACRs were relatively enriched for H3K9me2 in somatic tissues but depleted for active H3K4me3 and H3K27ac histone marks, with just over one-third (36.5%; 686 peaks) marked by H3K9me2 in leaves (*Figure 2B and C*; *Figure 2—figure supplement 1B*). Similarly, almost one-third of VN-specific ACRs significantly overlapped with a TE insertion (*Figure 2D and E*), with DNA/MuDR and LINE/L1 elements notably over-represented (*Figure 2F*). Accordingly, TEs specifically activated in pollen showed increased promoter accessibility specifically in the VN (*Figure 2G*; *He et al., 2019*). VN-specific ACRs were also significantly associated with regions that produce pollen siRNAs, including 21–22-nt class easiRNAs known to preferentially accumulate in pollen (*Figure 2C and D*; *Borges et al., 2018*; *Martinez et al., 2018*; *Slotkin et al., 2009*). These observations highlight how the VN gains chromatin accessibility within genomic regions normally silenced in sporophytic tissues, including TE-rich pericentromeric regions that lose constitutive heterochromatin, undergo transcription, and produce small non-coding RNAs.

## DME demethylates predicted TF binding sites in ACRs of the VN

The loss of constitutive heterochromatin in the VN is accompanied by selective DNA demethylation by DME, a DNA glycosylase that demethylates cytosines at a subset of genes flanked by transposons and repeats (*Choi et al., 2002*; *Schoft et al., 2011*). We thus explored how DME-dependent demethylation might impact chromatin accessibility in the VN. We first performed differential methylation analysis using published DNA methylomes to delimit hypermethylated cytosines in the VN of *dme-2/+* mutants compared to wild type (WT) (*Figure 3—figure supplement 1A–E*; *Supplementary file 2*; *Ibarra et al., 2012*). Strikingly, VN-specific ACRs were significantly enriched for regions demethylated by DME in the VN (*Figure 2C and D*), with just over half of all VN-specific ACRs (982 peaks; 52.3%) containing DME-dependent hypermethylated cytosines (*Figure 3A*; *Supplementary file 3*). A small fraction of common ACRs were also targeted for demethylation (2052 peaks; 12.8%) (*Figure 3B*; *Supplementary file 3*). Metaplots centred over DME-targeted ACRs illustrate the demethylation mediated by DME in all cytosine contexts of the VN (*Figure 3D*), which is mirrored by higher levels of chromatin accessibility in the VN than in SN (*Figure 3C*). Consistent with the VN-specific expression and activity of DME (*Park et al., 2017*), the levels of DNA methylation remained unchanged over these regions in WT and *dme-2/+* sperm. DME activity is thus unlikely to stimulate a gain in chromatin accessibility alone, at least at common ACRs, but rather modulates differential DNA methylation levels within these shared open chromatin sites. This suggests that chromatin accessibility likely precedes DME activity, consistent with the requirement of histone H1 depletion for DNA demethylation in the VN (*He et al., 2019*).

To further probe the role of DME, we performed motif enrichment analysis to determine which TFs are likely to be bound within DME-targeted ACRs. We used a set of experimentally derived DNA binding motifs generated by DAP-seq, an in vitro binding assay of the sequence specificity of TF-genomic DNA interactions (*O'Malley et al., 2016*). Fifty TFs expressed in the VN were present in the DAP-seq database, of which two-thirds (36 of 50; 72.0%) had significantly enriched motifs within DME-targeted ACRs (*Figure 3E*). These included motifs bound by MYB101 (*Figure 3E*), a pollen-specific TF that acts redundantly with MYB97 and MYB120 to regulate pollen tube growth (*Leydon et al., 2013*; *Liang et al., 2013*). The binding affinity of most TFs in Arabidopsis is strongly inhibited by DNA methylation, which was determined by performing the DAP-seq assay with methylated (DAP) or non-methylated (ampDAP) DNA (*O'Malley et al., 2016*). Of the 36 TFs predicted to bind within DME-targeted ACRs, 25 had been assessed in this manner (*O'Malley et al., 2016*), half of which (14 of 25; 56.0%) were inhibited by DNA methylation (*Figure 3F*). This suggests that DNA demethylation by DME facilitates the binding of several methylation-sensitive TFs within DME-targeted ACRs. Consistently, the predicted binding sites for these methylation-sensitive TFs were substantially hypo-methylated in VN from WT pollen (*Figure 3G–J*; *Figure 3—figure supplement 2A–J*). In *dme-2/+* mutant pollen, these predicted binding sites were hyper-methylated to around half of

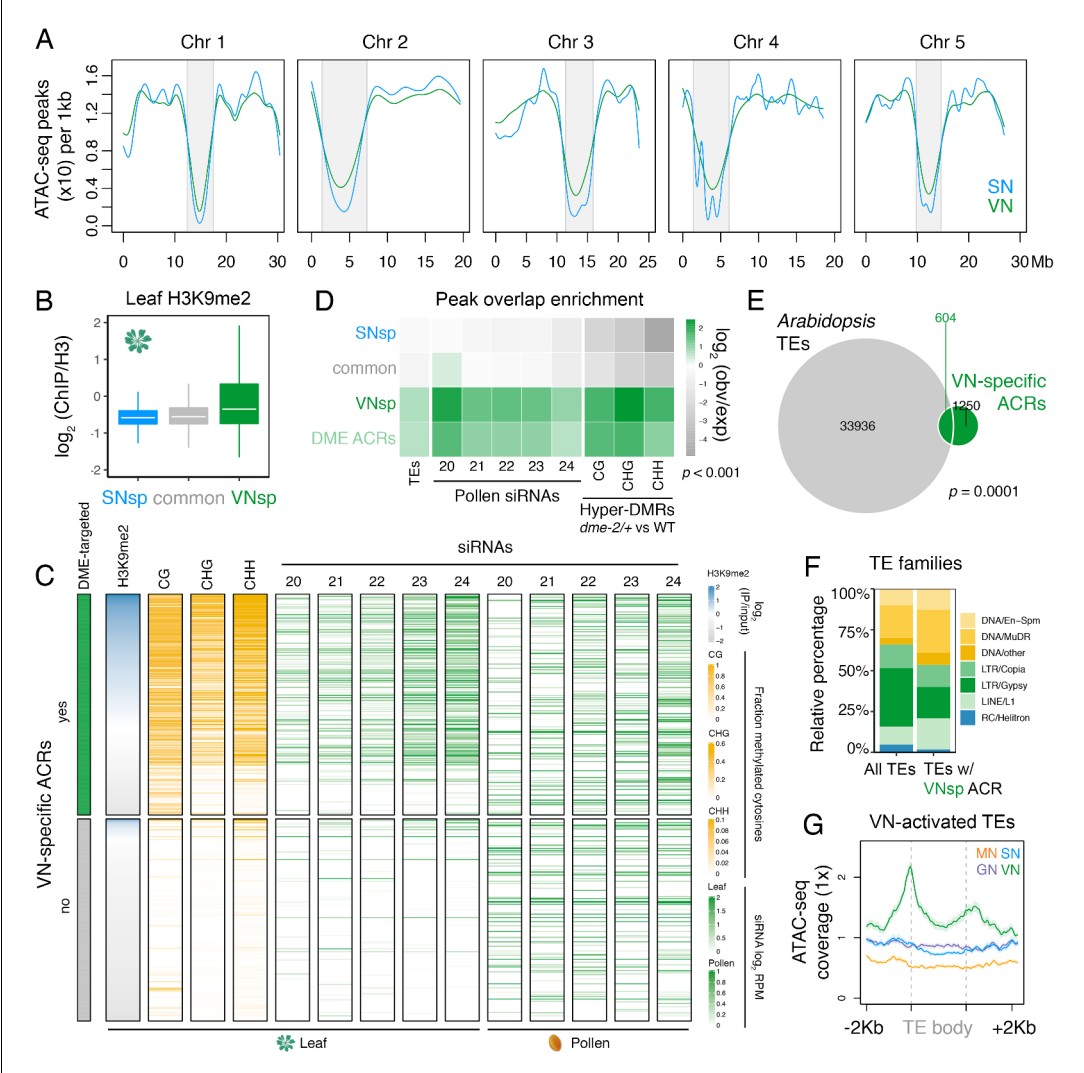

**Figure 2.** Pericentromeric sequences gain chromatin accessibility in the VN. (**A**) Distribution of accessible chromatin region (ACR) density over the five Arabidopsis chromosomes calculated in 10 kb bins. Pericentromeric regions are indicated with grey shading. (**B**) Levels of H3K9me2 marks in leaf tissue (log$_2$ ChIP-seq enrichment relative to H3) (*Baerenfaller et al., 2016*) at SN-specific, VN- specific, and common ACRs. Each boxplot indicates minimum and maximum values as well as 25th, 50th, and 75th quartiles. (**C**) Heatmap summarising the epigenetic state of VN-specific ACRs. Shown are the log$_2$ enrichment of H3K9me2 (relative to H3), the proportion of CG, CHH, and CHG methylation (*Stroud et al., 2013*) and different size classes of short-interfering RNAs (siRNAs) in leaves (*Papareddy et al., 2020*) alongside pollen siRNAs (*Borges et al., 2018*). VN-specific ACRs are grouped by the presence or absence of cytosines demethylated by DEMETER (DME) in the VN. (**D**) Heatmap summarising the overlap enrichment of SN-specific, VN-specific, DME-targeted, and common ACRs with TEs, pollen siRNAs (*Slotkin et al., 2009*), and *dme-2/+* hyper-DMRs. Fold changes were determined using hypergeometric tests compared with random Arabidopsis genomic regions (*n* = 10,000,000 permutations). (**E**) Pairwise overlap between VN-specific ACRs and Arabidopsis transposable elements (TEs). Significance of the enriched overlap (p-value) was determined in the hypergeometric test shown in panel C. (**F**) Distribution of TE gene classes associated with VN-specific ACRs alongside the relative frequency in the Arabidopsis genome. (**G**) Averaged ATAC-seq enrichment over TE genes specifically re-activated in the VN. Plotted is the ATAC-seq signal normalised to 1× Arabidopsis genome coverage.

The online version of this article includes the following figure supplement(s) for figure 2:

**Figure supplement 1.** DEMETER (DME) demethylates accessible chromatin uniformly across the VN.

the levels in WT sperm, consistent with the heterozygous nature of the *dme-2* mutant (*Figure 3G–J*; *Figure 3—figure supplement 2A–J*). These observations further delimit DME-mediated demethylation to motifs targeted by TFs expressed in the VN, which include several that are unable to bind efficiently in the presence of DNA methylation. In summary, we propose that a substantial portion of genomic regions that become preferentially or specifically accessible in the VN undergo DME-

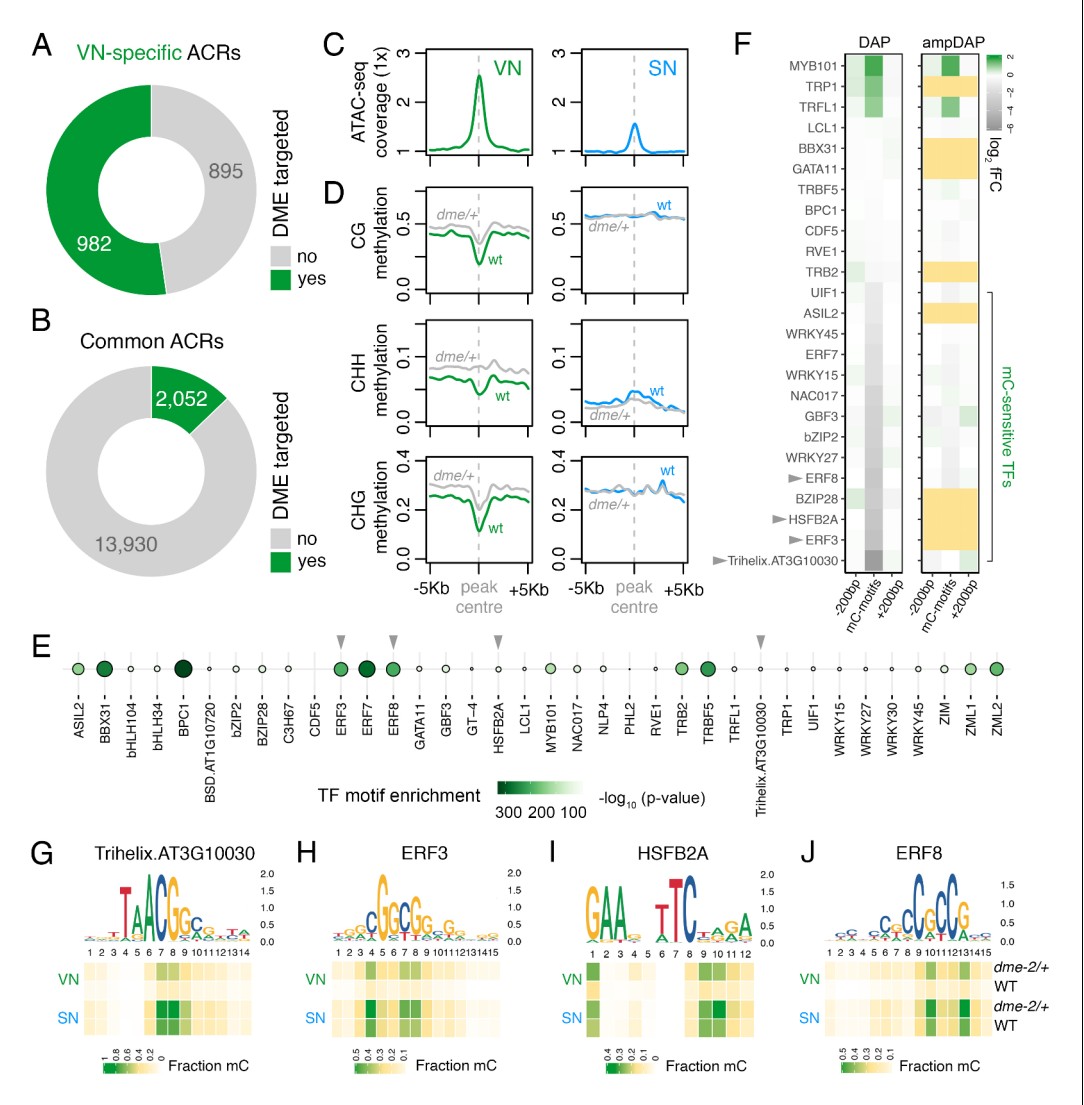

**Figure 3.** DEMETER (DME) demethylates predicted TF binding sites in accessible chromatin. (**A and B**) Relative proportion of VN-specific accessible chromatin regions (ACRs) (**A**) and common ACRs (**B**) that overlap with cytosines demethylated by DME in the VN. (**C and D**) Averaged metaplots comparing ATAC-seq enrichment (**C**) and the proportion of CG, CHH, and CHG methylation (**D**) in sperm (blue) and the VN (green) over DME-targeted ACRs. Regions with at least five differentially methylated cytosines were plotted. Coloured lines represent wild-type (WT) levels while grey lines indicate the level in nuclei isolated from *dme-2/+* mutant pollen. (**E**) Motifs of VN-expressed TFs enriched within DME-targeted ACRs. Significance (p-value) was assessed using a two-sided Fisher's exact test. TFs shown in panels **F–I** are marked with grey triangles. (**F**) Binding preference for the VN-expressed TF motifs enriched within DME-targeted ACRs. Left: DAP-seq binding fold-change at methylated motifs (mC-motifs) relative to neighbouring motifs (±200 bp). Right: ampDAP-seq binding fold-change at the same mC-motifs without DNA methylation. Yellow boxes indicate TFs without a score due to too few motifs. Data is from previously published DAP-seq experiments (***O'Malley et al., 2016***). TFs shown in panels **F–I** are marked with grey triangles. (**G–J**) DNA methylation level at predicted binding sites within DME-targeted ACRs for the four highest methylation-sensitive TFs shown in panel **E** – Trihelix.AT3G10030 (**G**), ERF3 (**H**), HSFB2A (**I**), and ERF8 (**J**). Each heatmap summarises the proportion of methylated cytosines over predicted binding sites in VN and SN of WT and *dme-2/+* pollen. Single nucleotides are indicated by the corresponding position frequency matrix logo plotted above each heatmap.

The online version of this article includes the following figure supplement(s) for figure 3:

**Figure supplement 1.** Differential methylated region (DMR) analysis between VN from wild-type (WT) and *dme-2/+* mutant pollen.

**Figure supplement 2.** DEMETER (DME) demethylates predicted TF binding sites specifically in the VN.

mediated DNA demethylation, which likely licenses the binding of several VN-expressed TFs involved in regulating the sporophyte-to-gametophyte transition.

## DNA demethylation by DME regulates the expression of pollen-tube related genes

DME-targeted ACRs clustered into two groups distinguished by a relative enrichment or depletion of H3K9me2, H3K27me1, and histone H1 in sporophytic tissue (*Figure 2—figure supplement 1C*). DME activity in open chromatin thus targets the genome uniformly in both pericentromeric regions and chromosome arms (*Figure 2—figure supplement 1D*). The 3034 DME-targeted ACRs were strongly enriched for TEs and for regions that produce siRNAs (*Figure 2C and D*), with around one-fifth (661 peaks, 21.8%) lying distally within intergenic regions (*Figure 4A*; *Supplementary file 3*). However, the majority of DME-targeted ACRs (2373 peaks, 78.2.8%) lied in the vicinity of protein-coding genes (*Figure 4A*; *Supplementary file 3*), which were significantly over-represented amongst VN-expressed genes (*Figure 4B*). Importantly, we observed a small but highly significant overlap between genes with a DME-targeted ACR and genes differentially expressed in *dme-2/+* mutant pollen (*Figure 4C and D*; *Figure 2—figure supplement 1E*), confirming that DME directly modulates gene expression during pollen development. These 27 genes included protein kinases, cysteine-rich peptides, metabolic proteins, as well as transcription factors, gene functions that are all known to be involved in pollen tube growth (*Figure 4D*; *Supplementary file 4*; *Supplementary file 5*; *Higashiyama and Takeuchi, 2015*; *Qu et al., 2015*).

The most significantly downregulated gene in *dme-2/+* pollen was a pollen-specific cysteine-rich *RECEPTOR-LIKE SERINE/THREONINE KINASE* (*RFK2*) (*Figure 4C*), which is embedded within H3K9me2-marked heterochromatin in the sporophyte (*Figure 4E*; *Figure 2—figure supplement 1C*). The upstream promoter region of *RFK2* was heavily hypomethylated in VN from WT pollen and contained several DME-demethylated cytosines including at predicted binding sites for bZIP2, ERF3, and ERF8 (*Figure 4E*), three DNA methylation-sensitive TFs (*Figure 3F*). Similarly, the downregulated AT2G16030 locus, which encodes a pollen-specific SAM-dependent methyltransferase also marked with sporophytic H3K9me2 (*Figure 2—figure supplement 1C*), contained several DME-demethylated cytosines at the predicted binding sites of ERF3, ERF7, ERF8 and a BSD domain TF (*Figure 4F*). Interestingly, DME-targeted ACRs were significantly associated with genes upregulated in growing pollen tubes, which occurs upon interaction of semi in vivo grown pollen tubes (SIVPT) with female pistil tissue (*Figure 4B*, SIVPT-induced) (*Leydon et al., 2017*; *Qin et al., 2009*). These 352 genes encode several examples of well-characterised pollen-specific receptor like-kinases, calcium-dependent protein kinases, cation/H+ exchanger proteins, and RALF-like signalling peptides (*Figure 2—figure supplement 1C*, grey labels) (*Myers et al., 2009*; *Sze et al., 2004*; *Takeuchi and Higashiyama, 2016*), around a quarter of which had little to no transcripts detectable in dry pollen (*Figure 2—figure supplement 1F*). Thus, DME demethylation acts to poise *cis*-regulatory regions for the onset of events taking place after pollination to facilitate the de novo wave of gene expression that occurs in the growing pollen tube (*Leydon et al., 2017*; *Qin et al., 2009*). In conclusion, chromatin accessibility in the VN strongly reflects the loss of constitutive heterochromatin, which facilitates DME-mediated demethylation of *cis*-regulatory elements predicted to be bound by methylation-sensitive TFs. VC differentiation is associated with these *cis*-regulatory regions, many of which are embedded within TE-rich regions of constitutive heterochromatin and inaccessible to transcription during sporophytic life.

## Accessible chromatin in sperm forecasts the sporophytic transition

In Arabidopsis, sperm chromatin is epigenetically reprogrammed through the global loss of H3K27me3, which involves the sperm-specific deposition of histone H3.10 and active demethylation by Jumonji-C family H3K27 demethylases *EARLY FLOWERING 6 (ELF6), RELATIVE OF ELF6 (REF6),* and *JUMONJI 13 (JMJ13)* (*Borg et al., 2020*). We thus explored how H3K27me3 reprogramming might impact chromatin accessibility in sperm. We previously showed that the global loss of H3K27me3 in sperm is accompanied by selective priming with or without H3K4me3 (*Borg et al., 2020*; *Figure 5A*). Consistently, H3K4me3-primed genes were marked by increased chromatin accessibility in their flanking promoter region and were enriched for SN-specific ACRs (*Figure 5A and B*, primed cluster). SN-specific ACRs were depleted for active marks in sporophytic

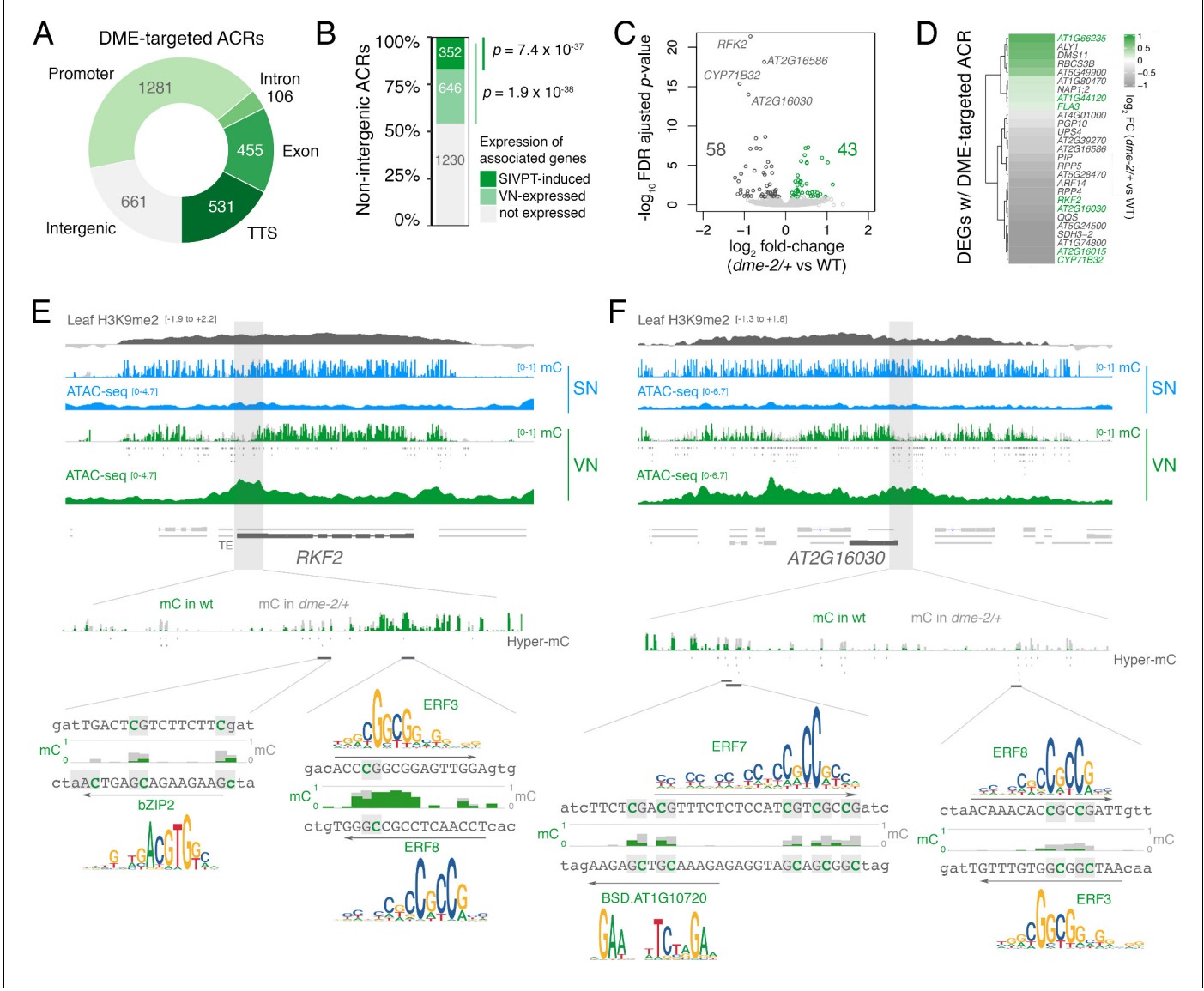

**Figure 4.** DEMETER (DME) activity regulates the expression of pollen tube-related genes. (**A**) Distribution of genomic features associated with DME-targeted accessible chromatin regions (ACRs). (**B**) Expression status of genes lying in the vicinity (<900 bp) of a DME-targeted ACR. Genes were classified as being expressed in the VN (TPM >1, light green) or by being upregulated during semi in vivo pollen tube (SIVPT) growth (dark green) (**Leydon et al., 2017**). Statistical enrichment of the overlap was determined using pairwise two-sided Fisher's exact tests. (**C**) Volcano plot of differentially expressed genes (DEGs) in *dme-2/+* pollen compared to wild type (WT). Upregulated genes are shown in green while downregulated genes are shown in dark grey. DEGs were defined as having an FDR-adjusted p-value <0.1. The top four most significantly downregulated genes are labelled. (**D**) Heatmap illustrating the magnitude of expression changes for DME-dependent DEGs directly associated with a DME-targeted ACR. Plotted is the log2 fold-change in *dme-2/+* pollen compared to WT. VN-specific genes are highlighted in green. (**E and F**) Browser view of two VN-specific direct DME target genes – *RECEPTOR-LIKE SERINE/THREONINE KINASE* (*RFK2*) (**E**) and the AT2G16030 locus encoding a SAM-dependent methyltransferase (**F**). Both genes are marked with H3K9me2 in the sporophyte (log2 ChIP-seq enrichment relative to H3). Tracks illustrate ATAC-seq signals and the proportion of methylated cytosines in all contexts (mC) in SN (blue) and VN (green), with DNA methylation levels in *dme-2/+* overlaid in grey. DME-dependent hyper-methylated cytosines (HMCs) are marked below. Shading indicates the flanking promoter region shown in the close-up view. Sequences surrounding predicted TF binding sites are shown, with the level of cytosine methylation at HMCs highlighted in green and grey for WT and *dme-2/+*, respectively. Arrows indicate the orientation and position of TF motifs with a corresponding position frequency matrix logo plotted above.

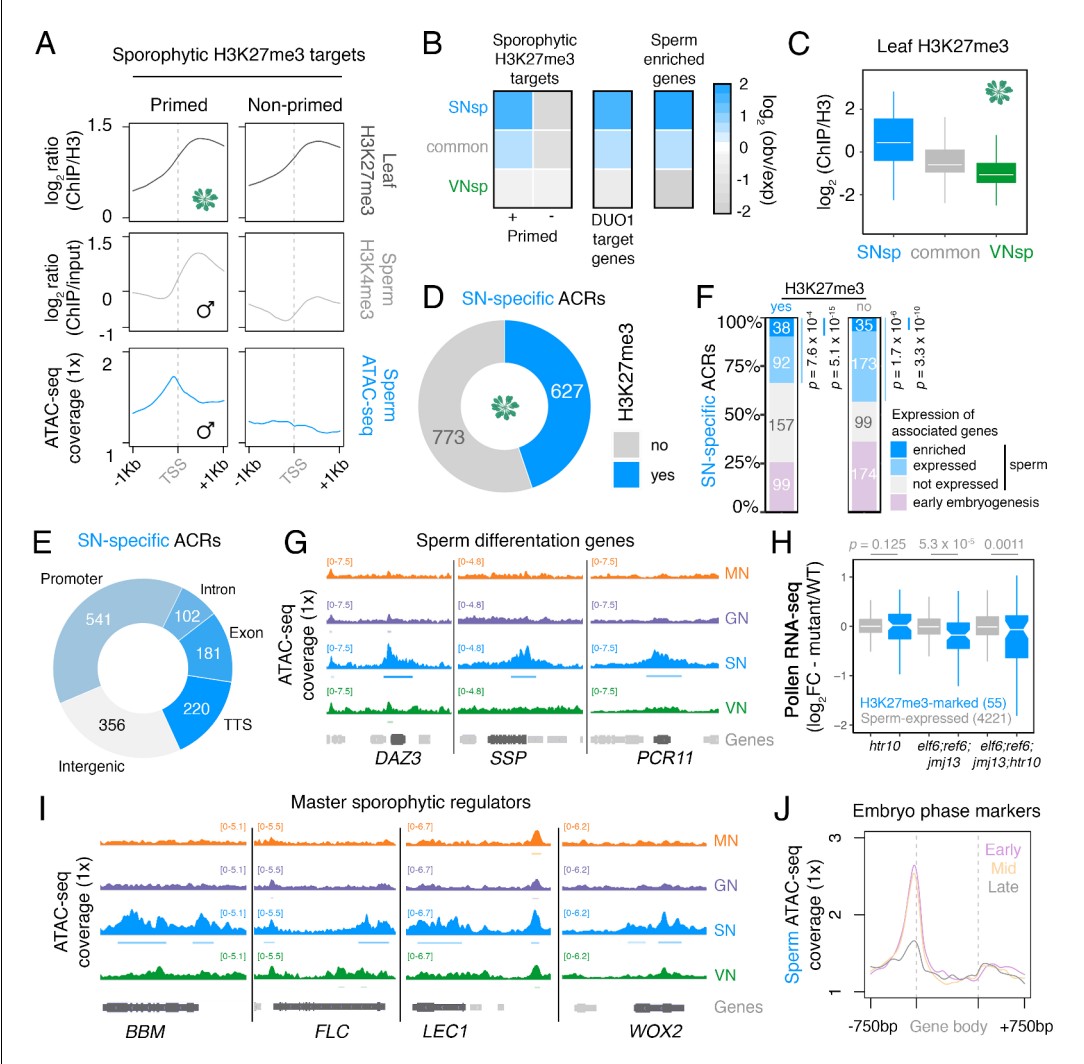

**Figure 5.** Chromatin accessibility in sperm forecasts sporophytic development. (**A**) Sperm ATAC-seq profiles reflect the reprogrammed state of sporophytic H3K27me3 target genes. Plotted are averaged levels of leaf H3K27me3 and sperm H3K4me3 (log$_2$ ChIP-seq enrichment relative to control) together with sperm ATAC-seq signals. ChIP-seq data and clustering for the presence and absence of H3K4me3 enrichment in sperm were described previously (**Borg et al., 2020**). (**B**) Heatmap summarising the overlap enrichment of SN-specific, VN-specific, and common accessible chromatin regions (ACRs) with somatic Polycomb target genes, DUO1 target genes, and sperm-enriched genes. Fold-enrichment was determined using hypergeometric tests compared with random Arabidopsis genomic regions (n = 10,000,000 permutations). (**C**) Levels of H3K27me3 marks in leaf tissue (log$_2$ ChIP-seq enrichment relative to H3) at SN-specific, VN-specific, and common ACRs. Each boxplot indicates minimum and maximum values as well as 25th, 50th, and 75th quartiles. (**D**) Relative proportion of SN-specific ACRs that overlap with a somatic H3K27me3 domain. (**E**) Distribution of genomic features associated with SN-specific ACRs. (**F**) Expression status of genes in the vicinity (<900 bp) of an SN-specific ACR with or without H3K27me3 in the sporophyte. Genes were classified as being expressed in sperm (TPM >1, light blue), by having enriched expression in sperm (dark blue), or by expression in the early zygote or embryos but not sperm (pink). Non-expressed genes in sperm are shown in grey. TPM >1 or <1 was used as a threshold for expressed and non-expressed genes, respectively. Statistical enrichment was determined using pairwise two-sided Fisher's exact tests. (**G**) Chromatin accessibility at three sperm-specific genes associated with sperm differentiation – *DUO1-ACTIVATED ZINC FINGER 1 (DAZ1), SHORT SUSPENSOR 1 (SSP1)* and *PLANT CADMIUM RESISTANCE 11 (PCR11)*. Tracks represent the ATAC-seq signal normalised to 1× Arabidopsis genome coverage for nuclei of the microspore (MN), germ cell (GN), sperm (SN), and vegetative cell (VN). (**H**) Differential expression between *htr10, elf6;ref6; jmj13*, and *elf6;ref6;jmj13;htr10* pollen relative to wild type (WT). Each boxplot indicates the minimum and maximum values as well as 25th, 50th, and 75th quartiles. Sample size (n) of genes associated with an H3K27me3-repressed ACR (blue) and all sperm-expressed genes (grey) is shown. Plot is restricted to genes with >10 counts in at least one RNA-seq sample and/or replicate. Statistical analysis was performed using two-sided Kolmogorov–Smirnov tests. (**I**) Chromatin accessibility at four major developmental regulators transcribed during early sporophyte development – *BABY BOOM (BBM), FLOWERING LOCUS C (FLC), LEAFY COTYLEDON 1 (LEC1)*, and *WUSCHEL-RELATED HOMEOBOX 2 (WOX2)*. Tracks represent the ATAC-seq signal normalised to 1× Arabidopsis genome coverage for nuclei of the microspore (MN), germ cell (GN), sperm (SN), and vegetative cell (VN). (**J**) Averaged ATAC-seq enrichment over genes with enriched expression during early (pink), mid (yellow), and late (grey) embryogenesis, which represent

*Figure 5 continued on next page*

*Figure 5 continued*

the pre-cotyledon phase, the transition phase, and the mature green phase, respectively (*Hofmann et al., 2019*). Plotted is the sperm ATAC-seq signal normalised to 1× Arabidopsis genome coverage.

leaf tissue but highly enriched for H3K27me3 (*Figure 5C*; *Figure 2—figure supplement 1B*). Almost one half of the SN-specific ACRs (627; 44.8%) overlapped a sporophytic H3K27me3 domain (*Figure 5D*; *Supplementary file 6*). The remaining SN-specific ACRs (773; 55.2%) lacked H3K27me3 in the sporophyte (*Figure 5D*; *Supplementary file 6*), suggesting sporophytic repression by an unknown pathway.

The majority of SN-specific ACRs (1044 peaks; 74.6%) were associated with a protein-coding gene (*Figure 5E*), around one-third of which were significantly over-represented for sperm-expressed genes (*Figure 5F*). These included several direct DUO1 target genes, including *DUO1-ACTIVATED ZINC FINGER (DAZ3)* and *PLANT CADMIUM RESISTANCE 11 (PCR11)*, as well as the paternally transcribed embryonic regulator *SHORT SUSPENSOR (SSP)* (*Figure 5G*; *Bayer et al., 2009*; *Borg et al., 2011*). The genes neighbouring H3K27me3-marked ACRs were significantly downregulated upon retention of H3K27me3, which occurs in mutant sperm compromised for the pathways that remove H3K27me3 (*Figure 5H*; *Borg et al., 2020*). Incidentally, H3K27me3 in GN remains comparable to levels in microspores but is specifically lost in divided sperm (*Borg et al., 2020*), further supporting our observation that major reorganisation of chromatin accessibility in sperm coincides with H3K27me3 reprogramming. These results reveal how H3K27me3 reprogramming exposes *cis*-regulatory elements to the transcriptional machinery during sperm specification.

In addition to genes transcribed in sperm, one quarter of the H3K27me3-repressed ACRs (110 peaks; 27.0%) were also associated with genes expressed specifically during the initiation of sporophytic development in the early embryo (*Figure 5F*). Strikingly, several master embryonic regulators all had high levels of chromatin accessibility in sperm, including *BABY BOOM (BBM)*, *LEAFY COTYLEDON 1 (LEC1)*, and *WUSCHEL-RELATED HOMEOBOX 2 (WOX2)*, as well as the floral repressor *FLOWERING LOCUS C (FLC)* (*Figure 5I*; *Boscá et al., 2011*; *Horstman et al., 2017*; *Tao et al., 2017*). This trend was broadly evident at genes with enriched expression during the early and middle phases of embryogenesis, which had higher levels of promoter accessibility than genes with enriched expression later during embryo maturation (*Figure 5J*; *Hofmann et al., 2019*). Thus, sperm chromatin is highly derived from the gametophytic state observed in the VN, since it is distinguished by accessible chromatin that forecasts the sporophytic transition in the next generation.

## Discussion

Here, we assess the landscape of accessible chromatin in the developing Arabidopsis male gametophyte and reveal major chromatin reprogramming at the transition between haploid and diploid life. Importantly, we show how the reprogramming of accessible chromatin and haploid gene expression is intricately linked with the differential erasure of distinct repressive epigenetic marks – H3K27me3 and H3K9me2-associated DNA methylation. Once microspore division is complete, the programmed removal of H3K9 and DNA methylation in the VC facilitates the expression of a fraction of the genes required for somatic haploid development. In parallel, H3K27me3 removal in sperm exposes *cis*-regulatory regions at developmental genes required to initiate transition back to diploid life after fertilisation. While substantial differences in ACRs have been observed across sporophytic Arabidopsis tissues (*Sullivan et al., 2014*), more closely related cell types appear to show few specific open chromatin sites (*Dorrity et al., 2020*; *Maher et al., 2018*). Despite being separated by a single cell division, thousands of differential ACRs distinguish the SN and VN, which is the result of the epigenetic reprogramming that occur across the male sporophyte-to-gametophyte transition. Future cell type-specific analysis of the sporophyte and female gametophyte promises to reveal whether the extent of transcriptional rewiring in the male haploid phase is shared during other major developmental transitions in the plant life cycle.

The epigenetic reprogramming events underlying the male haploid transition likely trace back to ancient, conserved mechanisms that accompany life form transitions in early land plants (*Ikeda et al., 2018*; *Okano et al., 2009*; *Schmid et al., 2018*; *Yaari et al., 2015*). Alternating haploid–diploid life forms in eukaryotes are likely derived from the dominant haploid phase of ancestral

unicellular organisms (*Niklas et al., 2014*). Multicellularity is proposed to have evolved in these haploid organisms first and was then followed by meiosis to mitigate spontaneous whole-genome duplications (*Lenormand et al., 2016*; *Niklas et al., 2014*; *Wilkins and Holliday, 2009*). Such events are likely to have initiated the first haploid–diploid life cycles during eukaryotic evolution (*Niklas et al., 2014*). Subsequent evolution of sexual reproduction through the innovation of gametes and fertilisation would have differentiated diploid development from that in the haploid phase (*Niklas et al., 2014*). In light of this, our findings provide insights into the molecular mechanisms that were likely adopted to control life form transitions during eukaryotic evolution. H3K27me3 likely evolved in unicellular eukaryotes prior to the repressive feedback loop that couples DNA with H3K9 methylation in land plants (*Bewick et al., 2017*; *Schuettengruber et al., 2017*). We propose that H3K27me3 reprogramming at the haploid-to-diploid transition might be a remnant of an ancestral role in the first haplo-diplo phasic life cycles. We also postulate that the DNA-H3K9 methylation feedback loop, which is normally associated with transposon silencing in flowering plants (*Feng and Michaels, 2015*), might have been co-opted to control diploid-to-haploid life form transitions during early land plant evolution. Importantly, our findings implicate H3K9 methylation in the transcriptional repression of not only transposons but also lineage specification in flowering plants, which has developmental relevance during early mouse organogenesis (*Nicetto et al., 2019*) but likely also in the specification of the female gametophyte (*Jiang et al., 2017*).

The vegetative cell represents the somatic cell of the male gametophyte and undergoes loss of constitutive heterochromatin, which facilitates active DNA demethylation by DME (*He et al., 2019*; *Mérai et al., 2014*; *Schoft et al., 2009*; *Schoft et al., 2011*). Chromatin accessibility in the VN supports the expression of TEs and easiRNAs caused by this epigenetic reconfiguration (*Ibarra et al., 2012*; *Slotkin et al., 2009*). Our findings suggest that the reprogramming and demethylation of pericentromeric heterochromatin are primarily required for VC differentiation. We identify several putative *cis*-regulatory elements actively demethylated by DME in the VN, many of which are silenced by constitutive heterochromatin in the sporophyte and are predicted to be bound by pollen-expressed TFs strongly repelled by DNA methylation (*O'Malley et al., 2016*). *DME* is essential for pollen viability and pollen tube germination in certain Arabidopsis ecotypes (*Schoft et al., 2011*), which is consistent with our findings that DME regulates the expression of several pollen tube-related genes. Intriguingly, the 3′ end of highly transcribed genes show prominent chromatin accessibility specifically in the VN. Physical proximity of the promoter and terminator through intragenic looping often facilitates rapid re-initiation of transcription (*Grzechnik et al., 2014*). Alternatively, these regions might be occupied by insulating proteins that prevent transcriptional read-through into neighbouring genes (*West et al., 2002*). Either mechanism would contribute to the high levels of transcription that are likely required to fuel rapid pollen tube growth (*Qin et al., 2009*; *Wang et al., 2008*).

DME-dependent demethylation preferentially targets short AT-rich TEs (*Ibarra et al., 2012*), suggesting that gametophytic development is at least partly specified through de-repression of TE-associated *cis*-regulatory sequences. Rewiring of gene expression by transposons and endogenous retroviruses is proposed to have contributed to the evolution and diversification of mammals (*Johnson, 2019*; *Lynch et al., 2011*). During early embryogenesis in mice and humans, accessible chromatin is widely shaped by fixed TE insertions that strongly associate with *cis*-regulatory elements (*Wu et al., 2016*; *Wu et al., 2018*). Novel TE and endovirus integrations can create novel promoters or TF binding sites that alter the regulation of nearby or even distant genes (*Rebollo et al., 2012*). We speculate whether TE integrations have imposed repression on the gametophytic programme during diploid life, which in turn could have facilitated the evolution of haploid–diploid life cycles in land plants. This hypothesis supports the general proposal that the fixation and selection of transposon-derived *cis*-regulatory elements marks specific periods of evolution within a lineage (*Jacques et al., 2013*; *Jordan et al., 2003*). In conclusion, our findings reveal the intricate relationship between epigenetic reprogramming and the regulatory epigenome during the alternation of generations in flowering plants.

## Materials and methods

### Plant materials and growth conditions

*Arabidopsis thaliana* seeds for ProHTR10:HTR10-Clover (*Kawashima et al., 2014*) and *dme-2/+* (*Choi et al., 2002*) were described previously. Plants were grown in long day (16 hr light/8 hr dark) conditions at 22°C.

### Plasmid construction and plant transformation

ProVCK:NTF and ProDUO1:MDB-NTF were generated using MultiSite Gateway Technology (Thermo Scientific). For the ProVCK:NTF construct, the NTF reporter with a stop codon (*Deal and Henikoff, 2011*) was recombined downstream of the vegetative cell-specific VCK promoter into the multisite gateway binary vector pB7m24GW,3 (VIB-UGent Gateway Vectors). The pDONRP4P1R-ProVCK entry clone was described previously (*Grant-Downton et al., 2013*). For the ProDUO1:MDB-NTF construct, the CYCB1;one mitotic destruction box (MDB) was fused in frame with the NTF reporter and driven with the DUO1 promoter in a strategy similar to that described previously (*Borg et al., 2014*). The pDONRP4P1R-ProDUO1 entry clone was described previously (*Brownfield et al., 2009*). The NTF cDNA was amplified from a ProGL2:NTF construct (*Deal and Henikoff, 2011*) using the primers NTF-attB1-F and NTF-attB2-R and cloned into pDONR221 or the primers NTF-attB2-F and NTF-attB3-R for cloning into pDONRP2RP3. The MDB cDNA without a stop codon was amplified with the primers MDB-attB1-F and MDB-attB2-R and cloned into pDONR221. The resulting binary vectors were transformed into a homozygous ProUBQ14:BirA marker line in Col-0 background.

The primer sequences were:

NTF-attB1-F GGGACAAGTTTGTACAAAAAAGCAGGCTCTATGGATCA
TTCAGCGAAAACCAC
NTF-attB2-R GGGGACCACTTTGTACAAGAAAGCTGGGTCTCAAGATCCACCAGTATCCTC
NTF-attB2-F GGGGCAGCTTTCTTGTACAAAGTGGCGATGGATCATTCAGCGAAAACC
NTF-attB3-R GGGGCAACTTTGTATAATAAAGTTGTTCAAGATCCACCAGTATCCTC
MDB-attB1-F GGGACAAGTTTGTACAAAAAAGCAGGCTTCATGATGACTTCTCGTTCGATTG
TTC
MDB-attB2-R GGGGACCACTTTGTACAAGAAAGCTGGGTCCTTCTCTCGAGCAGCAAC
TAAACC

### Fluorescence-activated cell sorting

Dry pollen from the ProVCK:NTF and ProHTR10:HTR10-Clover marker lines was harvested using a method published previously (*Johnson-Brousseau and McCormick, 2004*). SN and VN were isolated by FANS using a method described previously (*Borges et al., 2012*). Dry pollen was resuspended in ice-cold Galbraith Buffer (with 1% Triton X-100) (*Galbraith et al., 1983*) supplemented with 10 mM ß-mercaptoethanol and 1× complete protease inhibitor cocktail (Roche). After gentle rotation at room temperature for ~15 min, 500 µm glass beads were added to the pollen suspension and vortexed vigorously to disrupt pollen grains. Debris was removed by filtering the suspension through a 10 µm nylon mesh. The crude suspension was immediately subjected to FANS on a FACS Aria III cell sorter. The sorter was run in standard configuration, using a 70 µm ceramic nozzle with 1× PBS running at a constant pressure of 20 psi. A 488 nm blue laser was used to excite both GFP (i.e. NTF) and Clover and detected with a 530/30 nm bandpass filter.

Intact microspores were isolated by fluorescence-activated cell sorting (FACS) based on a method described previously (*Borges et al., 2012*). Unopened buds were picked from Col-0 wild-type plants and ground gently in ice-cold pollen extraction buffer (PEB: 10 mM $CaCl_2$, 2 mM MES, 1 mM KCl, 1% $H_3BO_3$, 10% Sucrose, pH 7.5). The crude spore preparation was then filtered through two layers of miracloth and the spore suspension centrifuged at 800 *g* for 5 min at 4°C to pellet the spores. The resulting pellet was resuspended in PEB, filtered through a 20 µm mesh, and immediately subjected to FACS on a FACS Aria III cell sorter. The sorter was run in standard configuration, using a 100 µm ceramic nozzle with 1× PBS running at a constant pressure of 20 psi. Microspores were gated for high angle scatter (SSC) and autofluorescence in the GFP channel using a 488 nm blue laser and 530/30 nm bandpass filter. Microspore purity and integrity were assessed by DAPI staining and microscopy immediately after FACS-based purification.

GN were isolated by FANS from a crude spore population isolated from the ProDUO1:MDB-NTF marker line. Unopened buds were picked from this line and ground gently in ice-cold 0.3 M mannitol solution. The crude spore preparation was filtered through a 100 µm nylon mesh and Triton X-100 added to a final concentration of 1%. The spore suspension was centrifuged at 800 *g* for 5 min at 4° C to pellet the spores. The resulting pellet was resuspended in ice-cold Galbraith Buffer (with 1% Triton X-100) (*Galbraith et al., 1983*) supplemented with 10 mM ß-mercaptoethanol and 1× complete protease inhibitor cocktail (Roche). 500 µm glass beads were added to the spore suspension and vortexed vigorously to disrupt the spores. Debris was removed by filtering the suspension through a 10 µm nylon mesh. The crude suspension was immediately subjected to FANS on a FACS Aria III cell sorter. The sorter was run in standard configuration, using a 70 µm ceramic nozzle with 1× PBS running at a constant pressure of 20 psi. A 488 nm blue laser was used to excite GFP (i.e. NTF) and detected with a 530/30 nm bandpass filter.

## RNA-seq analysis

For analysis of *dme-2/+* pollen, dry pollen was harvested from Col-*gl* and BASTA-sprayed *dme-2/+* plants using a method published previously (*Johnson-Brousseau and McCormick, 2004*). Pollen grains were disrupted in a Precellys tissue homogeniser (Bertin Instruments) and total RNA isolated using an RNeasy Micro Kit (Qiagen). For vegetative cell nuclei, FACS-isolated nuclei from the ProVCK:NTF line were sorted directly into Trizol reagent and precipitated with isopropanol. RNA-seq libraries were generated from the resulting RNA with Smart-seq2 (*Picelli et al., 2014*) using independent biological replicates. The libraries were sequenced on an Illumina Hiseq 2500 using 50 bp paired-end and single-end reads for VN and pollen samples, respectively.

Adapter trimming was performed using TrimGalore version 0.4.1 RRID:SCR_011847 (https://github.com/FelixKrueger/TrimGalore). Reads were aligned to the Arabidopsis genome (TAIR10) using the STAR aligner version 2.5.2a (*Dobin et al., 2013*). Transcripts per million (TPM) values were generated using Kallisto version 0.43.1 (*Bray et al., 2016*) with an index built on TAIR10 cDNA sequences (*Arabidopsis_thaliana.TAIR10.cdna.all.fa.gz*). Published RNA-seq data sets were analysed and described previously (*Borg et al., 2020*) with the addition of microspore RNA-seq data (*Wang et al., 2020*). Principal component analysis between different Arabidopsis tissues and cell types was based on the mean TPM value of corresponding biological replicates. Differential gene expression analysis between Col-*gl* and *dme-2/+* pollen was performed using DESeq2 version 1.22.2 (*Love et al., 2014*) for transcripts that had 10 counts or more in at least one sample. Differentially expressed genes (DEGs) were classified as having an FDR adjusted p-value <0.1. Overlap enrichment of DEGs with gene lists of interest was determined with the R package GeneOverlap 1.18.0 function *newGOM* (https://github.com/shenlab-sinai/GeneOverlap; RRID:SCR_018419).

## ATAC-seq analysis

For GN, SN, and VN ATAC-seq, ~50,000 FACS-purified nuclei were immediately resuspended in 25 µl transposase reaction mix (12.5 µl 2× TD buffer, 2.0 µl Nextera Tn5 [Illumina Cat #FC-121–1030], and nuclease-free water). For microspore ATAC-seq, FACS-purified spores were centrifuged at 800 *g* for 5 min at 4°C and the spore pellet resuspended in 500 µl lysis buffer (10 mM Tris-HCl, pH 7.4, 10 mM NaCl, 3 mM MgCl₂, 1% IGEPAL CA-630). The microspores were pelleted once more at 500 *g* for 10 min at 4°C and ~50,000,000 added to a 25 µl transposase reaction mix (12.5 µl 2× TD buffer, 2.0 µl Nextera Tn5 [Illumina Cat #FC-121–1030], and nuclease-free water). The Tn5 reaction mixes were incubated at 37°C for 1 hr and tagmented DNA fragments recovered with a MinElute PCR Purification Kit (Qiagen). Sequencing-ready libraries were generated as described previously (*Buenrostro et al., 2015*). The libraries were sequenced on an Illumina Hiseq 2500 using 50 bp paired-end reads. All ATAC-seq profiles were generated with at least two independent biological replicates.

Adapter trimming was performed with cutadapt version 1.9 using *–minimum-length 5 –overlap 1 –a* 'CTGTCTCTTATACACATCTCCGAGCCCACGAGAC' *-A* 'CTGTCTCTTATACACATCTGACGC TGCCGACGA' (https://github.com/marcelm/cutadapt/tree/v1.9.1; RRID:SCR_011841). Reads were mapped to the Arabidopsis genome (TAIR10) with Bowtie2 version 2.1.0 using *–end-to-end -X 2000 -x* (*Langmead and Salzberg, 2012*) and subsequently filtered for duplicate reads using Picard tools *MarkDuplicates* version 1.141 (https://github.com/broadinstitute/picard), RRID:SCR_006525. Only

reads with a mapQ >10 were used in our analysis. Biological replicates were subsequently merged for downstream analysis after confirming high correlation among replicates. Genome bigwig coverage files were generated with the deepTools version 2.2.4 utility *bamCoverage* using `–normali-zeTo1 × 119146348 –binSize=10 –smoothLength=100` (*Ramírez et al., 2014*). A cross-correlation matrix based on Pearson's correlation coefficient was generated by comparing bigwig coverage files of each replicate using deepTools version 2.5.0.1 utility *multiBigwigSummary* (*Ramírez et al., 2014*). Bigwig coverage files were visualised along the TAIR10 genome using IGV version 2.7.2 (*Robinson et al., 2011*). ATAC-seq peaks were called using the MACS2 version 2.1.0 *callpeak* function using *-B -q 1e-5 -f BAMPE* on merged replicate BAM files (*Zhang et al., 2008*). Annotation of ATAC-seq peaks to genes was performed using the R package ChIPseeker version 1.6.7 using Arabidopsis TxDb transcript metadata (*TxDb.Athaliana.BioMart.plantsmart28*). The region range to TSS was set to 900 bp (*Yu et al., 2015*), such that peaks >900 bp from the closest gene were considered as intergenic. Relative enrichment of the ATAC-seq peak groups to genome features was performed using the *annotatePeaks.pl* script in HOMER version 4.9 (*Benner et al., 2017*).

## Differential analysis of ATAC-seq peaks

For differential ATAC-seq peak analysis, all SN and VN peaks were combined and peaks lying within 50 bp of each other merged into one. Sporophyte-specific peaks were determined as microspore peaks that did not overlap with this combined set of peaks. This resulted in a master peak set with a total of 20,634 peaks. Reads mapping to the master peak set for each replicate were counted using the *featureCounts* function in Subread version 2.0.1 and used to generate FRiP scores. The resulting counts were used for pairwise differential analysis between the cell types using DESeq2 version 1.22.2 with default settings (*Love et al., 2014*). VN-specific and SN-specific peaks were classified as having an FDR adjusted p-value <0.05 and a $\log_2$ fold-change >1 or < −1, respectively. Common peaks were considered to have a $\log_2$ fold-change between 1 and −1.

## Analysis of ATAC-seq peak features

Chromosomal plots of ATAC-seq peak density based on the average number of peaks were calculated in 10 kb bins using bedtools *map* version 2.26 (https://bedtools.readthedocs.io/en/latest/). ATAC-seq heatmaps were generated using the R package EnrichedHeatmap (*Gu et al., 2018*). Averaged ATAC-seq profiles were generated using the EnrichedHeatmap function *normalizeToMatrix* and plotted using a custom script in R. Gene ontology (GO) enrichment was performed using the gProfileR package in R (*Reimand et al., 2011*) using gSCS correction, strong hierarchical filtering, and limited to biological process GO terms. Levels of histone marks associated with the ATAC-seq peaks were calculated with bedtools *map* version 2.26 (https://bedtools.readthedocs.io/en/latest/) using ChIP-seq coverage files analysed and described previously (*Borg et al., 2020*). Histone H1 ChIP-seq data was published previously (*Wollmann et al., 2017*). Peak overlap enrichment of ATAC-seq peak groups with TEs, siRNAs, and DMRs was performed with GAT using 10,000 permutations with random Arabidopsis genomic regions (https://gat.readthedocs.io/en/latest/).

## Small RNA re-analysis

Publicly available sRNA-seq reads obtained from *Slotkin et al., 2009* and *Borges et al., 2018* were analysed as described previously (*Papareddy et al., 2020*). First, sRNA reads were adaptor trimmed using Cutadapt v1.9.1 (https://github.com/marcelm/cutadapt/tree/v1.9.1). Trimmed sequences of 18–30 bp were aligned to the Arabidopsis genome (TAIR10) using STAR aligner version 2.7 (*Dobin et al., 2013*) requiring zero mismatches and ≤100 multiple end-to-end alignments. The resulting SAM files were used as input for *readmapIO.py* software (*Schon et al., 2018*) to reassign multimappers with a 'rich-get-richer' algorithm. The output bedFiles were sorted, condensed, and normalised for total genome matching reads. The level of siRNAs for VN-specific ACRs presented in the heatmap represent the mean reads per million (RPM) over the peaks including ±10 bp upstream and downstream.

## DNA methylation re-analysis

DNA methylation data sets were obtained from *Ibarra et al., 2012* and analysed as described previously (*Papareddy et al., 2020*). Briefly, sequenced reads were quality filtered and adaptor trimmed using TrimGalore version 0.6.2 (https://github.com/FelixKrueger/TrimGalore). Bisulphite-converted reads were aligned against the TAIR10 genome using Bismark version 0.22.2 with *bismark –non_directional -q –score-min L,0,–0.4* (*Krueger and Andrews, 2011*). BAM files containing clonal deduplicated and uniquely mapped reads were then used as substrate for Methylpy software version 1.2.9 (https://bitbucket.org/schultzmattd/methylpy) to extract weighted methylation rates at each cytosine as described previously (*Schultz et al., 2015*). Bisulphite conversion rates were calculated using the unmethylated chloroplast genome. Differentially methylated regions (DMRs) were defined using *Methylpy*. Differentially methylated cytosines (DMCs) with coverage $\geq 4$ overlapping reads were identified by root mean square tests and a false discovery rate $\leq 0.05$. Differentially methylated sites within 250 bp were collapsed into DMRs. DMRs were further filtered by discarding regions with <5 DMCs. Statistical analyses and associated figures were generated with a custom script in R. Sequence logo for DMCs was generated using the R package ggseqlogo (*Wagih, 2017*).

## Analysis of DME-targeted accessible chromatin

DME-targeted ACRs were defined as ATAC-seq peaks in the VN that overlapped with a hyper-DMC in *dme-2/+* VN compared to WT (see *DNA methylation analysis*). Overlap enrichment of genes associated with non-intergenic DME-targeted ACRs against gene lists of interest was determined with the R package GeneOverlap version 1.18.0 (https://github.com/shenlab-sinai/GeneOverlap). Motif enrichment was performed using motifs in the DAP-seq data set (*O'Malley et al., 2016*) for transcription factors with TPM >1 in the VN using *AME* in the MEME suite version 5.0.4 using all ATAC-seq peaks as a background model (*Bailey et al., 2009*). Binding $\log_2$ fold-change for these TFs in DAP-seq and ampDAP-seq experiments was described previously (*O'Malley et al., 2016*). Predicted TF binding sites within DME-targeted ACRs were determined using *FIMO* in the MEME suite version 5.0.4 with a FIMO threshold p-value <1e-5 and a background Markov model generated from all ATAC-seq peaks (*Bailey et al., 2009*). Pollen tube-induced genes were determined from previous semi in vivo pollen tube (SIVPT) data sets, with all genes in the intersection between SNP-based and microarray-based experiments considered (*Leydon et al., 2017*; *Qin et al., 2009*). ChIP-seq data for leaf H3K9me2 was described previously (*Baerenfaller et al., 2016*).

## Analysis of H3K27me3-repressed accessible chromatin

H3K27me3-repressed ACRs were defined as ATAC-seq peaks in SN that overlapped with somatic H3K27me3 domains described previously (*Borg et al., 2020*) using the R package ChIPpeakAnno function *findOverlapsOfPeaks* (*Zhu et al., 2010*). The clustering of somatic H3K27me3 target genes was based on H3K4me3 enrichment in sperm, with clusters 1 and 2 (primed) and cluster 3 (unprimed) described previously (*Borg et al., 2020*). Genome bigwig coverage files for somatic H3K27me3 and sperm H3K4me3 were also described previously (*Borg et al., 2020*). Overlap enrichment of genes associated with non-intergenic H3K27me3-repressed ACRs against gene lists of interest was determined with the R package GeneOverlap version 1.18.0 (https://github.com/shenlab-sinai/GeneOverlap). Peak overlap enrichment of ATAC-seq peak groups with different features was performed using GAT with 10,000 permutations of random Arabidopsis genomic regions (https://gat.readthedocs.io/en/latest/). DUO1 target genes were defined as all genes significantly induced by DUO1 overexpression in seedlings as described previously (*Borg et al., 2011*). Sperm-enriched genes were defined previously (*Borg et al., 2020*). Early embryogenesis genes were defined as genes with a TPM >1 in 14 hr zygotes or 2 cell embryos and a TPM <1 in sperm, using RNA-seq data described previously (*Borg et al., 2020*). Early and late embryo-enriched genes were defined previously and corresponded to pre-cotyledon and mature green phase markers, respectively (*Hofmann et al., 2019*).

## Acknowledgements

We thank Z Lorković and J M Watson for critical reading of the manuscript, S Akimcheva for guidance and technical support, and Yeonhee Choi for *dme-2* seeds. We thank the Next Generation

Sequencing and Plant Sciences facilities at the Vienna BioCenter Core Facilities GmbH, and the CLIP-CBE HPC team. This work was supported through core funding from the Gregor Mendel Institute, external grants from the FWF (P26887, I4258) and ERA-CAPS (EVO-REPRO I2163-B16). MB was supported through the FWF Lise Meitner fellowship M1818. RPK and MDN were supported by the European Research Council under the European Union's Horizon 2020 Research and Innovation Program (grant 637888 to MDN). DT was supported by Biotechnology and Biological Research Council funding (BB/I011269/1 and BB/N005090).

## Additional information

### Funding

| Funder | Grant reference number | Author |
|---|---|---|
| Austrian Science Fund | P26887 | Frédéric Berger |
| Austrian Science Fund | I 4258 | Frédéric Berger |
| Austrian Science Fund | I2163-B16 | Frédéric Berger |
| Austrian Science Fund | M1818 | Michael Borg |
| European Commission | ERC 637888 | Michael D Nodine |
| Biotechnology and Biological Sciences Research Council | BB/I011269/1 | David Twell |
| Biotechnology and Biological Sciences Research Council | BB/N005090 | David Twell |

The funders had no role in study design, data collection and interpretation, or the decision to submit the work for publication.

### Author contributions

Michael Borg, Conceptualization, Data curation, Software, Formal analysis, Funding acquisition, Validation, Investigation, Visualization, Methodology, Writing - original draft, Writing - review and editing; Ranjith K Papareddy, Rodolphe Dombey, Formal analysis; Elin Axelsson, Data curation, Software, Methodology; Michael D Nodine, David Twell, Funding acquisition; Frédéric Berger, Conceptualization, Formal analysis, Supervision, Funding acquisition, Validation, Investigation, Methodology, Project administration, Writing - review and editing

### Author ORCIDs

Michael Borg http://orcid.org/0000-0002-3982-3843
Rodolphe Dombey http://orcid.org/0000-0002-3670-4128
Elin Axelsson http://orcid.org/0000-0003-4382-1880
Frédéric Berger https://orcid.org/0000-0002-3609-8260

### Decision letter and Author response

Decision letter https://doi.org/10.7554/eLife.61894.sa1
Author response https://doi.org/10.7554/eLife.61894.sa2

## Additional files

### Supplementary files

• Supplementary file 1. Pollen ATAC-seq accessible chromatin regions (ACRs) detailed in this study.

• Supplementary file 2. Differentially hyper-methylated cytosines (DMCs) in vegetative cell nucleus (VN) from *dme-2/+* pollen compared to wild type (WT) described in this study.

• Supplementary file 3. DEMETER (DME)-targeted accessible chromatin regions (ACRs) described in this study.

• Supplementary file 4. Differential gene expression analysis of wild-type (WT) and *dme-2/+* pollen.

- Supplementary file 5. Pollen-tube related genes directly regulated by DEMETER (DME)-dependent demethylation.
- Supplementary file 6. H3K27me3-repressed accessible chromatin regions (ACRs) described in this study.
- Transparent reporting form

### Data availability

Deep-sequencing data that support the findings of this study have been deposited in the Gene Expression Omnibus (GEO) under accession code GSE155369. Re-analysis of previously published DNA methylomes from dme-2/+ pollen (Ibarra et al., 2012), and siRNAs from leaves (Papareddy et al., 2020) and pollen (Borges et al., 2018; Slotkin et al., 2009) were deposited in the GEO under accession code GSE155369.

The following dataset was generated:

| Author(s) | Year | Dataset title | Dataset URL | Database and Identifier |
|---|---|---|---|---|
| Borg M, Berger F | 2020 | Epigenetic reprogramming rewires transcription during the alternation of generations in Arabidopsis | https://www.ncbi.nlm.nih.gov/geo/query/acc.cgi?acc=GSE155369 | NCBI Gene Expression Omnibus, GSE155369 |

The following previously published datasets were used:

| Author(s) | Year | Dataset title | Dataset URL | Database and Identifier |
|---|---|---|---|---|
| Ibarra CA, Feng X, Schoft VK, Hsieh TF, Uzawa R, Rodrigues JA, Zemach A, Chumak N, Machlicova A, Nishimura T, Rojas D, Fischer RL, Tamaru H, Zilberman D | 2012 | Active DNA demethylation in plant companion cells reinforces transposon methylation in gametes | https://www.ncbi.nlm.nih.gov/geo/query/acc.cgi?acc=GSE38935 | NCBI Gene Expression Omnibus, GSE38935 |
| Borg M, Jacob Y, Susaki D, LeBlanc C | 2020 | Targeted reprogramming of H3K27me3 resets epigenetic memory in plant paternal chromatin | https://www.ncbi.nlm.nih.gov/geo/query/acc.cgi?acc=GSE120669 | NCBI Gene Expression Omnibus, GSE120669 |
| Borges F, Parent JS, van Ex F, Wolff P | 2018 | Transposon-derived small RNAs triggered by miR845 mediate genome dosage response in Arabidopsis | https://www.ncbi.nlm.nih.gov/geo/query/acc.cgi?acc=GSE106117 | NCBI Gene Expression Omnibus, GSE106117 |
| Slotkin RK, Vaughn M, Borges F, Tanurdžić M | 2009 | Epigenetic reprogramming and small RNA silencing of transposable elements in pollen | https://www.ncbi.nlm.nih.gov/geo/query/acc.cgi?acc=GSE61028 | NCBI Gene Expression Omnibus, GSE61028 |
| Papareddy RK, Páldi K, Paulraj S, Kao P | 2020 | Chromatin regulates expression of small RNAs to help maintain transposon methylome homeostasis in Arabidopsis | https://www.ncbi.nlm.nih.gov/geo/query/acc.cgi?acc=GSE152971 | NCBI Gene Expression Omnibus, GSE152971 |

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
