## [Decision Letter]

**Acceptance summary:**

Your paper describes the global epigenomic and transcriptomic changes that occur during Arabidopsis pollen development and provide a first glimpse of the genomic changes associated with the diploid-to-haploid transition in plants. This paper will be of particular interest to scientists working on plant reproduction, but also to a wider audience in the context of studies of sexual reproduction.

**Decision letter after peer review:**

Thank you for submitting your work entitled "Distinct epigenetic reprogramming events rewire transcription during life cycle transitions in Arabidopsis" for consideration by *eLife*. Your article has been reviewed by three peer reviewers, including Richard Amasino as the Reviewing Editor and Reviewer #1, and the evaluation has been overseen by Christian Hardtke as the Senior Editor.

The reviewers and Reviewing Editor have discussed the work with one another. Your paper is a high-quality study that addresses the important question of cell-type specific differences in gene expression using differential accessibility as a probe. However, there are some major issues that would need to be addressed for this work to be suitable for *eLife*.

One is to define the changes in chromatin accessibility during pollen development by including ATAC-seq data from bicellular microspores rather than focusing on comparisons with sister somatic cells.

The second is to perform chromatin accessibility assays in cells from dme2/+ pollen because the role of DME in rendering accessible the regulatory regions of genes that regulate pollen tube growth is currently based on correlations not supported by any experimental data.

The third is that the interpretations often extend beyond what is warranted by the data.

Below we provide more details on these issues, and we hope you find the efforts of the reviewers helpful.

As the editors have judged that your manuscript is of interest, but that additional experiments are required before it is published, we would like to draw your attention to changes in our revision policy that we have made in response to COVID-19 (https://elifesciences.org/articles/57162). First, because many researchers have temporarily lost access to the labs, we will give authors as much time as they need to submit revised manuscripts. We are also offering, if you choose, to post the manuscript to bioRxiv (if it is not already there) along with this decision letter and a formal designation that the manuscript is "in revision at *eLife*". Please let us know if you would like to pursue this option. (If your work is more suitable for medRxiv, you will need to post the preprint yourself, as the mechanisms for us to do so are still in development.)

Essential revisions:

1) The study is incomplete as the most critical stage in pollen development – the bicellular microspore stage (which encapsulates the initials of the two germ cell lineages), has not been included in this analysis. This is surprising considering that this stage is critical for male gamete and pollen development (as the authors also state in the manuscript), and that this experiment is technically feasible.

2) Comparing chromatin dynamics between two epidermal root cells and two pollen cells is not a valid argument to support the view that "haploid development" is not a result of cell differentiation. To address this point, authors should conduct a detailed analysis of all cell types at the three stages of pollen development.

3) The role of DME in poising accessible regulatory regions for the transcriptional events that regulate pollen tube growth should be validated experimentally. Authors should also perform chromatin accessibility assays in VN and SN using dme2/+ plants to confirm their hypothesis.

Similarly, the impact of DNA methylation at transcription factor binding sites that overlap with DME targeted ACRs should also be experimentally tested. Some of the correlations are weak and without experimental validation, their hypotheses are largely speculative.

4) The most significant comment is the conclusion that "epigenetic reprogramming" is deterministic instead of differentiation. The evidence to support this claim is that thousands of differential ACRs are identified in the VN, MN or the SN, but not between root hair and non-hair datasets. This is a strawman argument, that is built upon a negative result. Although it is true that the "hair" and "non-hair" datasets do not show differences in accessibility, it is likely a reflection of the lack of purity of these cells. Two recent single cell ATAC-seq studies using Arabidopsis somatic tissues were published on bioRxiv (Dorrity et al., 2020 and Farmer et al., 2020) and both show substantial variation of cell type specific differential accessibility. Furthermore, published studies comparing maize tissues from a couple of labs have shown thousands of regions of differential accessibility. Therefore, selecting to compare root hair and non-hair to support this major claim in the study is not advised. I believe focusing on what the data do show is actually significant enough for publication, without overinterpreting the results.

5) The evidence that regions that become accessible in the VN lose H3K9me2, siRNAs and DNA methylation requires additional analysis. Currently, the data are averaged for all regions. How many VN ACRs have H3k9me2, siRNAs, DNA methylation in somatic cells? What proportion of these actually lose H3k9me2, siRNAs and/or DNA methylation. One way to show this result is to take Figure 2C and create a heatmap where all the VN ACRs are rows and the columns are data for H3K9me2, CG, CHG, CHH, 20, 21, 22, 23 and 24 nt siRNAs. This would allow the reader to evaluate how many actual regions are enriched for these data.

6) Figure S2A does show high reproducibility, but correlations of 1.0 should not exist. These correlations show that read coverage per window is the same. If the windows are large enough the actual peaks do not have much influence over the result. For all replicates, please report the number of ACRs identified and their overlap of ACRs between each replicate. You could also consider using IDR.

7) The y-axis for enrichment in 2C is very small indicating enrichment is not as prevalent at these loci as the authors have presented.

8) The pollen data in 2G doesn't show enrichment of siRNAs. 20 nt siRNAs are not known to have a function, yet they are accumulating to greater levels than 22-24 nt siRNAs.

9) Evaluate if the siRNAs are siRNAs and not mRNA degradation products. This can be done be showing the siRNAs from individual regions are aligning to both strands of DNA instead of being derived from a single strand. This would explain the lack of enrichment of expected siRNA sizes and the over enrichment of classes that are not known to have function (20 and 25 nt siRNAs).

10) I really like the section of TF motif enrichment in these cell type specific ACRs. Given the purity and high-quality nature of the data, the authors should consider using DNA footprint analysis, which usually plagues bulk tissue data. It might provide more specificity that using the entire ACRs.

11) What are the negative controls used to identify TF motif enrichment in ACRs?

12) Figure 5A needs to control for gene length. Long genes will show more discrete H3K4me3, whereas short genes will short more enrichment over gene bodies.

13) The idea that chromatin accessibility in sperm is poised for sporophyte development has been previously proposed by the authors (Borg et al.,2020) but the data presented in this manuscript does not provide direct evidence to support this hypothesis.

[Editors' note: further revisions were suggested prior to acceptance, as described below.]

Thank you for submitting the revised manuscript of your work entitled "Distinct epigenetic reprogramming events rewire transcription during life cycle transitions in Arabidopsis" that addresses most of the issues raised in the first submission. There are a few more items from the review of your revised manuscript, that we would like for you and your co-authors to consider and respond to.

It would be a more useful contribution to literature if you would provide the Venn diagrams of ACRs between the individual samples prior to merging the replicate data. Your reason for not showing this was that the Spearman correlations were so high that it was unnecessary. However, adding the data for each replicate would permit the reader to appreciate the variability in the data. The addition of the FRIP scores in Figure S3 shows quite a bit of variation between samples. Although this doesn't invalidate the major conclusions of this study, the variability should be better acknowledged. Showing overlap of ACRs from individual replicates with and between samples is one way to do so.

The addition of the heatmap to Figure 2 is much appreciated. However, it shows that the majority of ACRs do not possess H3K9me2 and/or siRNAs in leaf tissue. This result does not invalidate conclusions, but this is important to note in the main text. In particular, note how many ACRs overlap and H3K9me2 region and make it clear to the reader that only a minor number of ACRs in the VN are affected.

The conclusion that there are thousands of differential ACRs during this developmental progression as compared to somatic tissues based on published ATAC-seq data is not likely to hold up over time. Published studies are limited in their data analyses preventing accurate identification of cell-type-specific ACRs. The limits of other published studies are in contrast to your work and perhaps you do not want to appear critical of other studies, but your work would be a better contribution to the field if it was made clear that the major rewiring of cis-regulatory elements that you have observed in the haploid phase of the life cycle may very well occur in other phases of development when cell-type-specific methods and more sophisticated data analyses are applied.

---

## [Author Response]

Essential revisions:1) The study is incomplete as the most critical stage in pollen development – the bicellular microspore stage (which encapsulates the initials of the two germ cell lineages), has not been included in this analysis. This is surprising considering that this stage is critical for male gamete and pollen development (as the authors also state in the manuscript), and that this experiment is technically feasible.

We thank the reviewers for this suggestion. The bicellular pollen stage encapsulates the stage after microspore division, where the germ (or generative) cell becomes engulfed in the cytoplasm of its sister vegetative cell. The germ cell is present during a very narrow and transient window during pollen development, with no specific molecular markers available. Bicellular pollen remains embedded within unopened floral buds, making this a technically very challenging developmental stage to isolate, and it is not a routine procedure as suggested by the reviewers.

To address this, we generated a tailored marker line that specifically and transiently marks germ cell nuclei using a ProDUO1:MDB-NTF construct. NTF is a nuclear envelope targeted GFP marker, while MDB denotes the CYCB1;1 destruction box. The construct is driven with the DUO1 promoter, which is active solely in germ cells and sperm cells (Borg et al., 2014). MDB-NTF is susceptible to degradation by APC/C outside of the G2/M-phase (Borg et al., 2014). See Figure 1—figure supplement 1H for details. In pollen at mid-bicellular stage, the GFP signal in the germ cell nucleus shows signal similar to autofluorescence background in the pollen cell wall. In late bicellular pollen, germ cells enter late G2 phase and MDB-NTF accumulate to high levels well-above background autofluorescence. As germ cells exit mitosis, APC/C is reactivated leading to progressive turnover of the NTF marker in the resulting two sperm, which only show a cytoplasmic signal barely above autofluorescence. We devised sorting parameters that allowed to isolate germ cell nuclei from pollen at late bicellular stage (Figure 1—figure supplement 1I).

We isolated germ cell nuclei from this marker line to perform ATAC-seq profiling (Figure 1—figure supplement 3A-B). Like microspores, we observed no clear nucleosome laddering (Figure 1—figure supplement 3C). Nonetheless, we observed strong overlap with other ATAC-seq libraries, suggesting it is still useable to support the major interpretations of this study. Our analysis did not reveal de novo gains in chromatin accessibility in germ cells compared with microspores but rather a gradual loss in accessibility among the class of sporophytic-specific peaks (Figure 1C and D). Incidentally, H3K27me3 in germ cell nuclei also remains comparable to levels in microspores (Borg et al., 2020) but is specifically lost in divided sperm, further supporting our observation that major reorganization of chromatin accessibility in sperm coincides with H3K27me3 reprogramming.

2) Comparing chromatin dynamics between two epidermal root cells and two pollen cells is not a valid argument to support the view that "haploid development" is not a result of cell differentiation. To address this point, authors should conduct a detailed analysis of all cell types at the three stages of pollen development.

This was a nice suggestion from the reviewers and as requested, we have performed pairwise comparisons (now described in Figure 1—figure supplement 3D) between the four stages of pollen development we profiled. Because the SN and VN represent the culmination of chromatin accessibility reprogramming in pollen, we focused on the ACRs recovered from the SN vs VN comparison together with a set of sporophyte-specific ACRs. These 20,634 ACRs represent the master peak set in our pairwise analyses.

These comparisons reveal the progressive accumulation of differential ACRs during pollen development, which remain modest in more closely related cell types (ex. MSP vs GN, GN vs SN) but reach a maximum at the latest stage of pollen development (i.e. between the SN and VN). These data are now presented in a new main figure panel (Figure 1F).

We realize that some confusion has arisen in regard to the differences we highlight between haploid and diploid development. We show that extensive chromatin accessibility changes in pollen are a result of unique and major waves of epigenetic reprogramming that only occur in the haploid gametophyte generation. Thus, 1000s of regulatory regions that are inactive and repressed in somatic tissues lose repressive marks in pollen, and in turn become active and gain chromatin accessibility. It thus stands to reason that these reprogramming events serve to establish and distinguish the haploid gametophytic program from the diploid program. Importantly, the genome-wide loss of repressive marks occurs after a single cell division, which in turn allows for extensive and rapid rewiring of transcriptional activity. Nonetheless, we agree that our discussion point might have led to the impression that "haploid development is not a result of cell differentiation*”*. Please see the response to major comment 4 for more clarification on these points.

3) The role of DME in poising accessible regulatory regions for the transcriptional events that regulate pollen tube growth should be validated experimentally. Authors should also perform chromatin accessibility assays in VN and SN using dme2/+ plants to confirm their hypothesis. Similarly, the impact of DNA methylation at transcription factor binding sites that overlap with DME targeted ACRs should also be experimentally tested. Some of the correlations are weak and without experimental validation, their hypotheses are largely speculative.

The reviewers comment that the impact of DNA methylation at the TF binding sites should be tested experimentally. in vitro assays have already been performed in the DAP-seq dataset used in our analysis (O’Malley et al., 2016). Additional approaches would require ChIP-seq analysis of TF binding in pollen, or in vitro in gel shift assays with methylated and unmethylated DNA. We believe that such experiments extend far beyond the aims of this study, which focuses on delivering a genomic level perspective of how chromatin accessibility is reprogrammed during the alternation of generations.

To validate the DME-targeted ACRs, we performed RNA-seq profiling of *dme-2^+/-^* pollen and could confirm a direct association with at least 27 of the genes mis-regulated in *dme-2* – this is highly significant and statistically supported (Figure 2—figure supplement 1E; *p* = 2.6 x 10^-8^) – even when working within the limitations of using *dme-2^+/-^* plants. We doubt that in the context of our analysis presented in Figure 3 and Figure 4, ATAC-seq profiling of *dme-2^+/-^* would be more insightful than the RNA-seq profiling we have already performed. Isolating *dme* mutant pollen from WT pollen produced by *dme-2^+/-^* plants would also require tailored fluorescent marker lines that as-yet do not exist. Moreover, the use of SYBR-green dye to isolate the VN and SN from a mixed population (which is a common approach to separate the two cells) is not recommended for ATAC-seq analysis, since DNA intercalating dyes are known to strongly disrupt chromatin structure (please see https://kb.10xgenomics.com/hc/en-us/articles/360027640311-Can-I-sort-nuclei-for-Single-Cell-ATAC-sequencing-or-Single-Cell-Multiome-ATAC-GEX-).

4) The most significant comment is the conclusion that "epigenetic reprogramming" is deterministic instead of differentiation. The evidence to support this claim is that thousands of differential ACRs are identified in the VN, MN or the SN, but not between root hair and non-hair datasets. This is a strawman argument, that is built upon a negative result. Although it is true that the "hair" and "non-hair" datasets do not show differences in accessibility, it is likely a reflection of the lack of purity of these cells. Two recent single cell ATAC-seq studies using Arabidopsis somatic tissues were published on bioRxiv (Dorrity et al., 2020 and Farmer et al., 2020) and both show substantial variation of cell type specific differential accessibility. Furthermore, published studies comparing maize tissues from a couple of labs have shown thousands of regions of differential accessibility. Therefore, selecting to compare root hair and non-hair to support this major claim in the study is not advised. I believe focusing on what the data do show is actually significant enough for publication, without overinterpreting the results.

We understand the concerns raised. As we clarified in our response to point 3, our reasoning is that epigenetic reprogramming is deterministic for the establishment of the haploid gametophyte programs, which in turn facilitates sperm and vegetative cell differentiation. These two processes are clearly inter-connected, and we do not imply that reprogramming is deterministic *instead of* differentiation.

Despite being separated by what is essentially a single cell division, we observe 1000s of unique ACRs between the sperm and vegetative cell. We have performed a similar comparison with sister cells of root epidermis to emphasize the scale of these changes. During the review process, we were referred to recent papers of scATAC-seq profiling of *Arabidopsis* root cells. We quote findings from Dorrity et al., 2020 below:

"For each cell type, the median number of genes with tissue-specific accessibility was 20 (range 5 to 53) (Figure 1C). This small number of genes is consistent with earlier studies that show few open chromatin sites that define cell type identity in *A. thaliana*.^7, 23^ Although thousands of differentially accessible sites have been found across tissue types,**^7^**accessibility differences between more closely related cell types remains largely unexplored, with the exception of root hair vs non-hair, in which very few differences were found.^**7**,**11**^ "

Thus, the findings from Dorrity et al., appear to support our reasoning since differentiation between closely-related sporophytic cell types involves less than 50 ACRs. This contrasts strongly with the 1000s of changes in the haploid generation between two closely-related cell types – which importantly are only separated by a single cell division. We have now revised this section by removing the pairwise comparison between the root epidermis cell types and toned down our conclusions on this point.

5) The evidence that regions that become accessible in the VN lose H3K9me2, siRNAs and DNA methylation requires additional analysis. Currently, the data are averaged for all regions. How many VN ACRs have H3k9me2, siRNAs, DNA methylation in somatic cells? What proportion of these actually lose H3k9me2, siRNAs and/or DNA methylation. One way to show this result is to take Figure 2C and create a heatmap where all the VN ACRs are rows and the columns are data for H3K9me2, CG, CHG, CHH, 20, 21, 22, 23 and 24 nt siRNAs. This would allow the reader to evaluate how many actual regions are enriched for these data.

We are very thankful for this useful suggestion from the reviewers. As requested, we have added a new heatmap to Figure 2 that summarizes the levels of H3K9me2, DNA methylation and siRNAs in sporophytic leaf tissue for all VN-specific ACRs. This heatmap has now replaced the bar chart previously occupying Figure 2G. The heatmap is grouped by whether the ACRs are targeted by DME or not. The heatmap nicely illustrates how VN-specific ACRs are highly enriched for H3K9me2 and DNA methylation in the sporophyte, as well as the stimulation of pollen siRNAs from regions that are not observed in sporophytic tissue. This is consistent with our peak overlap enrichment test shown in Figure 2C. This adds further evidence that the VN-specific gain of accessible chromatin in the gametophyte generation stimulates transcription of genomic regions normally silenced in sporophytic tissues.

6) Figure S2A does show high reproducibility, but correlations of 1.0 should not exist. These correlations show that read coverage per window is the same. If the windows are large enough the actual peaks do not have much influence over the result.

We thank the reviewer for pointing out this very valid point. After some consideration, we realize that a Spearman correlation is likely more appropriate here since ATAC-seq data by definition is not normally distributed. We have thus re-analysed the correlation between our datasets alongside our newly added germ cell nucleus profiles and report a Spearman correlation matrix (Figure 1—figure supplement 3B).

For all replicates, please report the number of ACRs identified and their overlap of ACRs between each replicate. You could also consider using IDR.

Because our ATAC-seq replicates showed such high genome-wide correlation (Figure 1—figure supplement 3B), we merged replicates for our differential analysis of ATAC-seq profiles. We reasoned that this would increase read complexity and improve peak detection, with which we then generated a master peak set for pairwise comparisons. This was then used in a DESeq2 comparison with the individual replicates to statistically assess chromatin accessibility within these regions – this has now been clarified with a new schematic diagram (Figure 1—figure supplement 3D).

7) The y-axis for enrichment in 2C is very small indicating enrichment is not as prevalent at these loci as the authors have presented.

We agree and have improved the heatmap legend to present a more gradual gradient between the green and grey colors. These values represent the log_2_ fold-change of observed overlaps for our ACR peaks groups against those expected from 10,000 permuted overlaps with random genomic regions. The positive enrichment values (in green), which are only observed for VN-specific ACRs, all range between a log_2_ fold-change of 0.6 to 2.1 – the inverse log of which equates to 1.6x to 5.5x the frequency of overlaps expected by chance. Importantly, all of the overlap tests we performed were statistically significant, as indicated by *p* < 0.001 below the heatmap legend. In contrast, common and SN-specific ACRs were all significantly depleted for these features, highlighting the association of VN-specific ACRs with heterochromatic features known to be altered during pollen development. We hope that our explanation clarifies this point.

8) The pollen data in 2G doesn't show enrichment of siRNAs. 20 nt siRNAs are not known to have a function, yet they are accumulating to greater levels than 22-24 nt siRNAs.9) Evaluate if the siRNAs are siRNAs and not mRNA degradation products. This can be done be showing the siRNAs from individual regions are aligning to both strands of DNA instead of being derived from a single strand. This would explain the lack of enrichment of expected siRNA sizes and the over enrichment of classes that are not known to have function (20 and 25 nt siRNAs).

We thank reviewers for raising this point. We have replaced previous panel Figure 2G with a new heatmap in Figure 2C, which assesses the mean level of siRNAs over the ACRs including ±10bp upstream and downstream. As requested, we have also assessed the strand bias for 20nt, 21-22nt and 23-24nt siRNAs. As the plot in Author response image 1 shows, pollen siRNAs align equally to both the + and – strand, confirming that all size classes are not simply mRNA degradation products. We have not included this quality control plot as part of the revised manuscript since we are using already published and robustly analysed pollen siRNA data (Borges et al., 2018). The reviewer is correct in saying that no clear function has been assigned to 20nt siRNAs. However, the specific enrichment of 20-22nt siRNAs in pollen has already been reported, with 21-22nt siRNAs having been shown to regulate paternal genome dosage in *Arabidopsis* (Martinez et al., 2018). We have corrected our statement in the revised version to refer to 21-22nt siRNAs to avoid confusion and cited this important study.

10) I really like the section of TF motif enrichment in these cell type specific ACRs. Given the purity and high-quality nature of the data, the authors should consider using DNA footprint analysis, which usually plagues bulk tissue data. It might provide more specificity that using the entire ACRs.

We thank the reviewer for their positive comment about our analysis of TF motif enrichment. As suggested, we have performed DNA footprinting analysis with HINT-ATAC, which assesses TF activity through bias corrected cleavage analysis. Increased HINT-ATAC signals over TF motifs implies increased TF activity in a cell (Li et al., 2019), which was evident for several TFs enriched within DME-targeted ACRs, including the methylation-sensitive TFs (Figure 3K-N; Figure 3—figure supplement 2). This suggests active TF occupancy at the predicted binding sites that become accessible through the epigenetic reprogramming of constitutive heterochromatin in the VN.

11) What are the negative controls used to identify TF motif enrichment in ACRs?

We thank the reviewer for pointing out this omission. We now mention that to perform motif enrichment, we used the complete set of pollen ACRs recovered in our study as negative control for the Analysis of Motif Enrichment (AME) tool in MEME.

12) Figure 5A needs to control for gene length. Long genes will show more discrete H3K4me3, whereas short genes will short more enrichment over gene bodies.

We appreciate this as a highly valid point since the averaging we performed had been scaled to a consistent length over the whole gene body. The metaplots in Figure 5A have been replotted and are now averaged upstream and downstream of the transcription start site to control for gene body length. This has been simplified into two clusters – one with and without H3K4me3 (i.e primed and un-primed), with the primed cluster specifically characterized by increased promoter accessibility that is followed by H3K4me3 accumulation.

13) The idea that chromatin accessibility in sperm is poised for sporophyte development has been previously proposed by the authors (Borg et al.,2020) but the data presented in this manuscript does not provide direct evidence to support this hypothesis.

We use the term “poised” to refer to the transcriptionally-primed states that we report in sperm chromatin, which specifically targets genes expressed during the earliest phases of embryogenesis and which is corroborated with both chromatin accessibility (this study) and active H3K4me3 marks (Borg et al. 2020). We have revised our use of the term “poised” and replaced this with “primed” or “forecast” throughout the revised manuscript. How sperm chromatin state influences transcriptional events after fertilization are indeed very interesting questions. Building on these important observations with further studies would allow one to explore the direct transgenerational impacts of reprogramming and any potential paternal effects. These would be such novel findings that they necessitate further investigations in the framework of a new and in-depth study. Such studies would require highly technical isolation of very early zygotes for gene expression profiling, which is by no means a straightforward experiment and would require new collaborations. We hope that the reviewers realize and agree that addressing these points go far beyond the scope of the present study, which reports the first example of how transcription is reprogrammed during the alternation of generations, and for which we show compelling evidence that chromatin accessibility, and by extension the binding of TFs to active *cis*-regulatory regions, is largely regulated through epigenetic reprogramming of H3K27me3 in sperm.

[Editors' note: further revisions were suggested prior to acceptance, as described below.]

1) It would be a more useful contribution to literature if you would provide the Venn diagrams of ACRs between the individual samples prior to merging the replicate data. Your reason for not showing this was that the Spearman correlations were so high that it was unnecessary. However, adding the data for each replicate would permit the reader to appreciate the variability in the data. The addition of the FRIP scores in Figure S3 shows quite a bit of variation between samples. Although this doesn't invalidate the major conclusions of this study, the variability should be better acknowledged. Showing overlap of ACRs from individual replicates with and between samples is one way to do so.

As requested, we have now added pie charts of ACR overlaps between among the replicates for each sample. This is indeed a useful addition to help the reader identify any variability, which we acknowledge is present in our data. As clarified in our revised manuscript, we control for this by merging replicates prior to peak calling, then using the replicates later to statistically assess differential chromatin accessibility within these regions using DESeq2.

2) The addition of the heatmap to Figure 2 is much appreciated. However, it shows that the majority of ACRs do not possess H3K9me2 and/or siRNAs in leaf tissue. This result does not invalidate conclusions, but this is important to note in the main text. In particular, note how many ACRs overlap and H3K9me2 region and make it clear to the reader that only a minor number of ACRs in the VN are affected.

As requested, we now explicitly state the number of VNsp ACRs that overlap with H3K9me2 as follows: “just over one-third (36.5%; 686 peaks) are marked by leaf H3K9me2”. This proportion is not minor, but we acknowledge that selection of VC expressed genes is not only the product of removal of marks of constitutive heterochromatin in the discussion “Once microspore division is complete, the programmed removal of H3K9 and DNA methylation in the VC facilitates the expression of a fraction of the genes required for somatic haploid development.”

3) The conclusion that there are thousands of differential ACRs during this developmental progression as compared to somatic tissues based on published ATAC-seq data is not likely to hold up over time. Published studies are limited in their data analyses preventing accurate identification of cell-type-specific ACRs. The limits of other published studies are in contrast to your work and perhaps you do not want to appear critical of other studies, but your work would be a better contribution to the field if it was made clear that the major rewiring of cis-regulatory elements that you have observed in the haploid phase of the life cycle may very well occur in other phases of development when cell-type-specific methods and more sophisticated data analyses are applied.

We have moved our small discussion points from the Results section to the beginning paragraph of the Discussion section and added the following sentence: “Future cell type-specific analysis of the sporophyte and female gametophyte promises to reveal whether the extent of transcriptional rewiring in the male haploid phase is shared during other major developmental transitions in the plant life cycle.” Still, we maintain that our data shows that transitions between life forms are the product of epigenetic mechanisms distinct in scale and identity from cell differentiation